# Space Filling Curves as Spatial Priors for Small or Data-Scarce Vision Transformers

## Abstract

Vision Transformers (ViTs) have become a dominant backbone in computer vision, yet their attention mechanism lacks inherent spatial inductive biases, which are especially crucial in small models and low-data regimes. Inspired by the masking in Linear Transformers and the scanning patterns of Vision SSMs, we propose VIOLIN [1], a lightweight masked attention mechanism that integrates *Space Filling Curves (SFCs)* to enhance spatial awareness with negligible computational overhead. VIOLIN scans the input image with multiple SFCs to build curve specific decay masks, which are averaged and multiplied with the attention matrix to encode spatial relationships. It yields notable gains in data-scarce settings: when fine-tuning on VTAB-1K, VIOLIN improves accuracy by up to $8.7\%$ on the Structured group, and it can be combined with parameter-efficient tuning methods such as LoRA. Beyond fine-tuning, VIOLIN consistently improves various tiny or small scale ViT architectures (e.g., DeiT, DINO) during pretraining, achieving gains of up to $0.9\%$ on on ImageNet-1K and $7.2\%$ on pixel level CIFAR-100. Overall, VIOLIN offers a computationally efficient yet effective way to inject spatial inductive bias into ViTs, particularly benefiting small models and data-scarce scenarios.

 Anonymous VIOLIN Code

## 1 Introduction

Vision Transformers (ViTs) (Dosovitskiy et al., 2021) have rapidly become a dominant architecture in computer vision, achieving strong performance across diverse tasks. Their success comes from capturing global dependencies through self-attention, but unlike Convolutional Neural Networks (CNNs) (O'Shea & Nash, 2015), ViTs lack inherent *spatial priors* such as locality (Fan et al., 2024). This makes them highly *data-hungry* and *dependent on larger model sizes*.[2] While sufficient parameters and massive datasets allow ViTs to learn these biases directly (Lu et al., 2022; Sun et al., 2017), many downstream tasks require adapting a pretrained backbone with limited data. In such cases, *even large ViTs struggle to specialize*, making stronger inductive biases essential across scales. Prior works have attempted to address this limitation with convolutions (Guo et al., 2022), novel positional encodings (Wu et al., 2021b), or masking strategies (Fan et al., 2024).

Concurrently, in natural language processing, State Space Models (SSMs) and Linear Transformers have emerged as efficient alternatives to standard transformers (Gu & Dao, 2024; Dao & Gu, 2024; Sun et al., 2023), and their vision adaptations have achieved strong results (Alkin et al., 2024; Liu et al., 2024b; Zhu et al., 2024). Through recurrence and a decay factor on attention scores, these models can capture the relative spatial order of image patches. However, this information depends entirely on the chosen scanning order, and to capture both vertical and horizontal relations, they typically require multiple directional scans (Li et al., 2024).

Scanning an image converts its 2D patch layout into a 1D sequence, with the order of patches determined by a traversal path. This process can be viewed as a Space Filling Curve (SFC): a continuous path that passes through every point in a multidimensional grid while systematically covering the entire image (Sagan, 1994). Many vision backbones, including vanilla ViT (Dosovitskiy et al., 2021), Vision x-LSTM (Alkin et al., 2024), VMamba (Liu et al., 2024b), and Vim (Zhu et al.,

---

[1]As a subtle homage to Giuseppe Peano, the creator of space filling curves, we named our model in a way that also reflects a musical instrument, just like Peanos family name resembles Piano.

[2]We define models with $\leq$ 30M parameters as small-scale and those with $\sim$86M+ as large-scale.

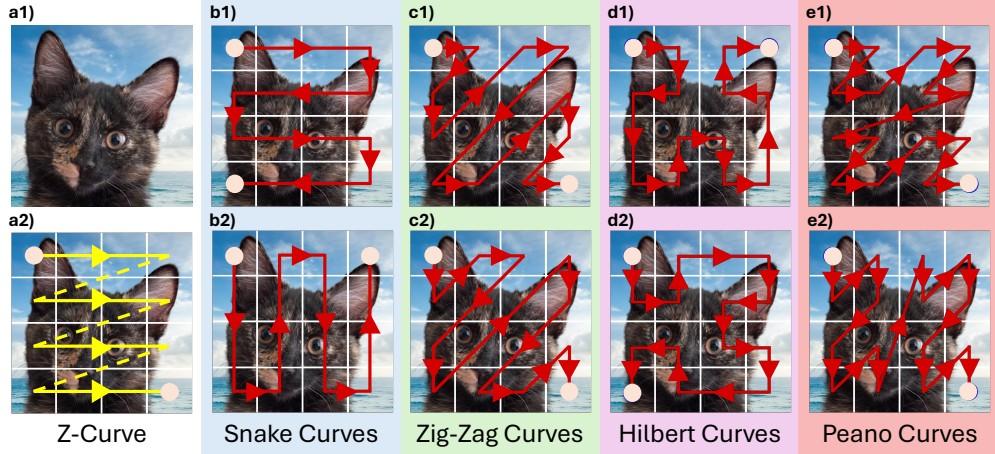

Figure 1: *Space Filling Curve paths:* Examples of traversal paths used in VIOLIN on a $4 \times 4$ patched image. **(a1)** Original image. **(a2)** Z-curve **(b1)** Snake curve, **(b2)** Transposed Snake curve, **(c1)** Zig-zag curve, **(c2)** Transposed Zig-zag curve, **(d1)** Hilbert curve, **(d2)** Transposed Hilbert curve, **(e1)** Peano curve, **(e2)** Transposed Peano curve.

2024), use the simple Z-curve, or row-by-row scan, for this linearization (see Figure 1 **(a2)**). Given that other SFCs, such as Snake, Zig-zag, Peano, and Hilbert curves, preserve locality in different ways (see Figure 1), we ask the following question:

*Can incorporating SFC-inspired structure into attention help to enhance the spatial understanding of ViTs, thereby improving their performance in small models and limited-data settings?*

In this work, we answer this question affirmatively by proposing VIOLIN , a lightweight attention mechanism that injects spatial priors through SFC-guided decay masks. Specifically, VIOLIN integrates multiple SFC based scans into a single decay mask, $\mathbf{M}_{\text{VIOLIN}}$ , which captures the relative spatial locations of image patches *without altering the rest of the architecture*. As a result, VIOLIN provides an efficient, plug-and-play way to introduce locality into ViTs, *particularly benefiting small models and data-scarce scenarios*. Figure 1 **(b - e)** shows the SFCs used in VIOLIN to traverse the image, with their corresponding linearized sequences presented in Figure 10.

We evaluate VIOLIN across a broad set of settings:

- Fine-tuning DeiT, DeiT-III, and DINO (Touvron et al., 2021; 2022; Caron et al., 2021) on VTAB (Zhai et al., 2019), across scales *from Tiny (5M) to Huge (632M)*, where VIOLIN consistently improves baselines with gains up to **8.7%** on individual tasks and **4.7%** on average. VIOLIN can also be seamlessly combined with parameter-efficient fine-tuning methods, further boosting adaptability.

- Pretraining small-scale models on ImageNet-1K (Russakovsky et al., 2015), where VIOLIN improves performance by up to **0.9%**, and on pixel-level CIFAR-100 (Krizhevsky, 2009), achieving a striking **7.2%** improvement.

- Additional analyses, including the complementary contributions of different curves, performance on the Structured VTAB category, and extensions to dense prediction tasks such as object detection on COCO (Lin et al., 2015) and semantic segmentation on ADE20K (Zhou et al., 2017), further highlight the versatility of VIOLIN and the importance of explicitly modeling spatial priors.

## 2 BACKGROUND

**Notations and preliminaries** We denote a patched image as $\mathcal{I} \in \mathbb{R}^{H \times W \times d}$, where $H$ and $W$ are the number of patches along height and width, and $d$ is the embedding dimension. Its flattened form is $\mathbf{X} \in \mathbb{R}^{N \times d}$ with $N = H \times W$ as the sequence length. For single head attention, the query, key, and value matrices $\mathbf{Q}, \mathbf{K}, \mathbf{V} \in \mathbb{R}^{N \times d}$ are computed using learnable weights $\mathbf{W}_Q, \mathbf{W}_K, \mathbf{W}_V \in \mathbb{R}^{d \times d}$,

and the standard ViT attention is computed as

$$\mathbf{Q} = \mathbf{X}\mathbf{W}_Q, \ \mathbf{K} = \mathbf{X}\mathbf{W}_K, \ \mathbf{V} = \mathbf{X}\mathbf{W}_V, \qquad \mathbf{Y} = \text{Softmax}\left(\frac{\mathbf{Q}\mathbf{K}^\top}{\sqrt{d}}\right)\mathbf{V}. \qquad (1)$$

where $\mathbf{Y} \in \mathbb{R}^{N \times d}$ is the attention output. We use $h$ and $L$ for the number of attention heads and transformer layers respectively. Elements of matrices and vectors are accessed by $[\cdot]$, and $\odot$ denotes the Hadamard product. A full list of notations is provided in Appendix A.

**Vision Transformers and spatial priors**    After dividing an image into patches (tokens), ViTs process them as a 1D sequence, typically flattened with a Z-curve (Dosovitskiy et al., 2021; Touvron et al., 2021; Caron et al., 2021), as shown in Figure 1 **(a2)**, which discards information about neighboring patches. To reintroduce spatial information, most ViTs add positional embeddings before transformer blocks, where self-attention captures token interactions. Recent works have further improved performance through self-supervised learning (e.g., DINO (Caron et al., 2021; Oquab et al., 2023)) and optimized training strategies (e.g., DeiT and DeiT-III (Touvron et al., 2021; 2022)). In this study, we show how VIOLIN improves upon these models and training recipes.

By processing patches independently, ViTs lack the strong spatial inductive bias of architectures like CNNs, which inherently encode locality (Yuan et al., 2021). Although ViTs capture global interactions, they struggle with fine-grained local structures, making training data-hungry (d'Ascoli et al., 2021). Sufficiently large models and datasets can mitigate this by learning locality from data, but when model size or data is limited, ViTs struggle to achieve strong performance (Lu et al., 2022), see Appendix B.1 for details.

**Linear Transformers**    Linear attention was introduced as an alternative to softmax attention, reducing quadratic complexity to linear time via a recurrent formulation eq. (2) (Katharopoulos et al., 2020). Instead of relying on positional embeddings to capture the order within a sequence, most modern Linear Transformers (Sun et al., 2023) incorporate a decay factor ($\gamma$),

$$\mathbf{S}_i = \gamma \mathbf{S}_{i-1} + \mathbf{k}_i^\top \mathbf{v}_i, \quad \mathbf{y}_i = \mathbf{q}_i^\top \mathbf{S}_i \quad \Leftrightarrow \quad \mathbf{Y} = (\mathbf{Q}\mathbf{K}^\top \odot \mathbf{M}_{\text{Causal}})\mathbf{V}, \quad \mathbf{M}_{\text{Causal}}[i,j] = \begin{cases} \gamma^{i-j} & i \ge j, \\ 0 & i < j. \end{cases} \quad (2)$$

where $\mathbf{S}_i \in \mathbb{R}^{d \times d}$ is the hidden state. This recurrent form can be parallelized using matrix multiplication with a Toeplitz decay mask $\mathbf{M}$ (Qin et al., 2023; Sun et al., 2023). Though linear masked attention was initially proposed for causal NLP tasks, it is later adapted to non-causal tasks using full Toeplitz masks (Afzal et al., 2025). The decay mask naturally extends context length, supports variable sequence lengths, and provides locality information that inspired VIOLIN .

**Scans in Linear Vision Transformers and SSMs**    Linear Transformers and SSMs have been applied to vision tasks (Alkin et al., 2024; Liu et al., 2024b; Zhu et al., 2024). To enhance spatial representation, these models often traverse image patches using a Z-curve, typically scanning in both vertical and horizontal directions. Each scan acts as a separate recurrence, capturing distinct spatial patterns through their own decay factors.

**Space Filling Curves**

**Definition 2.1.** A *Space Filling Curve (SFC)* is a continuous mapping from a closed unit interval $S = [0,1]$ to a closed unit hypercube $Q = [0,1]^N$, passing through every point in $Q$ exactly once (Peano, 1990). In this work, we focus on the 2D Euclidean case $Q = [0,1]^2$, corresponding to the image domain.

Based on definition 2.1, many SFCs can been defined, including the **Snake**, **Peano** (also known as the Morton curve) (Peano, 1990), **Hilbert** (Hilbert, 1935), **Z** (or Sweep), and **Zig-zag** (Wallace, 1992) curves as illustrated in Figure 1. Additionally, other curves include the Sierpiski (Sierpiski, 1915), Lebesgue (Lebesgue, 1904), and Schoenberg curves (Schoenberg, 1938).

Flattening or scanning can be viewed as applying an SFC $c$ to a 2D patched image $\mathcal{I}$ with $N$ total patches, mapping it into a 1D sequence $\mathbf{X}_c \in \mathbb{R}^N$ via a flattening function $F_c(\mathcal{I}) : \mathbb{R}^{H \times W} \to \mathbb{R}^N$

$$F_c(i,j) : (i,j) \mapsto n, \ i \in \{0, \dots, H-1\}, \ j \in \{0, \dots, W-1\}, \ n \in \{0, \dots, N-1\}, \quad (3)$$

$$\mathbf{X}_c = F_c(\mathcal{I}), \quad \mathbf{X}_c[n] = \mathcal{I}[i,j] \ \text{where} \ n = F_c(i,j). \quad (4)$$

This flattening can be applied independently across each embedding dimension $d$ for $\mathcal{I} \in \mathbb{R}^{H \times W \times d}$. While SFCs have diverse applications in other domains, their role in image classification remains underexplored (Zhao et al., 2024; Kutscher et al., 2025). For further details, please refer to Appendix B.3.

## 3 METHODOLOGY

In this section, we first introduce decay-masked attention in Section 3.1, then extend it to capture diverse scanning patterns in Sections 3.2 and 3.3, and finally formulate VIOLIN attention in Section 3.4.

### 3.1 ATTENTION WITH DECAY MASK

As shown in Appendix C.1, attention (eq. (1)) is permutation equivariant. In other words, changing the order of tokens in the sequence results in the same reordering in the output. Therefore, standard attention does not encode relative spatial priors within an image. To introduce locality, we take inspiration from Linear Transformers and multiply a decay mask with the attention:

$$\mathbf{Y} = \text{Softmax}\left(\frac{\mathbf{Q}\mathbf{K}^\top}{\sqrt{d}} \odot \mathbf{M}\right)\mathbf{V}, \quad \mathbf{M}[i,j] = \gamma^{|i-j|}, \, 0 < \gamma \leq 1. \tag{5}$$

This decay mask $\mathbf{M}$, also known as the KacMurdockSzeg matrix (Kac et al., 1953), extends the causal decay mask to full attention (Sun et al., 2023; Afzal et al., 2025). It dampens the attention score between tokens $i$ and $j$ by $\gamma^{|i-j|}$, enforcing locality in the flattened sequence $\mathbf{X}$. However, both the token order in $\mathbf{X}$ and the notion of distance in $\mathbf{M}$ depend entirely on how the original image $\mathcal{I}$ is flattened. This raises a natural question: *What are alternative, principled ways to flatten an image?*

### 3.2 SFCs AS PRINCIPLED WAY OF IMAGE FLATTENING

Following eq. (4), scanning an image along a path $c$ yields the sequence $\mathbf{X}_c = F_c(\mathcal{I})$. Many ViTs (Dosovitskiy et al., 2021; Touvron et al., 2021) use the Z-curve as the default scanning method.

**Z-Curve** The Z-curve, also called sweep, row-major order, or raster scan, traverses the image row by rowtop to bottom, and left to right within each row. Its flattening function is $F_z(i,j) = iW + j$. See Appendix B.3 for details on curves used in this study.

Although flattening with different curves usually requires reprocessing the image, we propose a simpler and significantly more efficient alternative: *applying a permutation to the flattened sequence.*

**Permutation of a flattened image** Given a sequence $\mathbf{X}_{c_1}$ flattened via SFC $c_1$, and noting that flattening is one-to-one, we define a permutation $\pi_{c_1 \to c_2} : \{0, \ldots, N-1\} \to \{0, \ldots, N-1\}$ that maps it to $\mathbf{X}_{c_2}$ from curve $c_2$

$$\mathbf{X}_{c_2} = \pi_{c_1 \to c_2}(\mathbf{X}_{c_1}). \tag{6}$$

Note that since each index in $\mathbf{X}_{c_1}$ uniquely corresponds to one in $\mathbf{X}_{c_2}$, $\pi_{c_1 \to c_2}$ is invertible. Alternatively, we can represent it as a permutation matrix $\mathbf{P}_{c_1 \to c_2} \in \{0, 1\}^{N \times N}$

$$\mathbf{P}_{c_1 \to c_2}[n,m] = \begin{cases} 1 & \text{if } m = \pi_{c_1 \to c_2}(n), \\ 0 & \text{otherwise,} \end{cases} \quad \mathbf{X}_{c_2} = \mathbf{P}_{c_1 \to c_2}\mathbf{X}_{c_1}. \tag{7}$$

Since $\mathbf{P}_{c_1 \to c_2}$ is a permutation matrix, $\mathbf{P}_{c_2 \to c_1} = \mathbf{P}_{c_1 \to c_2}^{-1} = \mathbf{P}_{c_1 \to c_2}^\top$. Thus, by flattening the image once using the Z-curve, it is possible to obtain $\mathbf{X}_c$ for other curves by applying $\pi_{z \to c}(\cdot)$.

### 3.3 SFCs MEET ATTENTION

With the naive approach, using $\mathbf{X}_c$ for each curve individually and following eq. (5), the output of masked attention $\mathbf{Y}_c$ can be calculated such that

$$\mathbf{Y}_c = \text{Softmax}\left(\frac{\mathbf{Q}_c\mathbf{K}_c^\top}{\sqrt{d}} \odot \mathbf{M}_c\right)\mathbf{V}_c, \quad \text{where} \quad \mathbf{M}_c[i,j] = \gamma_c^{|i-j|}, \tag{8}$$

where $\mathbf{Q}_c, \mathbf{K}_c, \mathbf{V}_c$ are the corresponding query, key, and value matrices. Note that as the token order of $\mathbf{Y}_c$ depends on the curve $c$, when multiple curves are used, the outputs (e.g $\mathbf{Y}_{c_1}$ and $\mathbf{Y}_{c_2}$) will have mismatched positions. To overcome this issue we can define a basis for our curves as below.

**Basis Curve** After computing the attention output $\mathbf{Y}_c$ for each curve $c$, we permute them into a common basis order to align all outputs. This preserves each curves spatial locality while ensuring they share a consistent reference order. Following standard ViT flattening, we use the Z-curve as the basis and perform all permutations relative to it, simplifying notation as $\pi_{z\to c} = \pi_c$, $\pi_{c\to z} = \pi_c^{-1}$ and $\mathbf{P}_{z\to c} = \mathbf{P}_c$, $\mathbf{P}_{c\to z} = \mathbf{P}_c^{-1}$. The output aligned to the basis is

$$\widetilde{\mathbf{Y}}_c = \pi_c^{-1}(\mathbf{Y}_c) = \mathbf{P}_c^\top \mathbf{Y}_c. \tag{9}$$

**Permutation of Decay Mask** The aligned output $\widetilde{\mathbf{Y}}_c$ of the masked attention in eq. (8) is

$$\widetilde{\mathbf{Y}}_c = \mathbf{P}_c^\top \mathbf{Y}_c = \mathbf{P}_c^\top \mathrm{Softmax}\left(\frac{\mathbf{Q}_c \mathbf{K}_c^\top}{\sqrt{d}} \odot \mathbf{M}_c\right) \mathbf{V}_c. \tag{10}$$

Equivalently, we can permute the decay mask $\mathbf{M}_c$ to the basis order as $\widetilde{\mathbf{M}}_c = \pi_c^{-1}(\mathbf{M}_c) = \mathbf{P}_c^\top \mathbf{M}_c \mathbf{P}_c$, allowing attention to be computed directly in the basis, see Section C.3 for proof. The attention output then becomes

$$\widetilde{\mathbf{Y}}_c = \mathrm{Softmax}\left(\frac{\mathbf{Q}\mathbf{K}^\top}{\sqrt{d}} \odot \widetilde{\mathbf{M}}_c\right) \mathbf{V}, \quad \widetilde{\mathbf{M}}_c = \pi_c^{-1}(\mathbf{M}_c), \ \mathbf{M}_c[i,j] = \gamma_c^{|i-j|}. \tag{11}$$

This approach is more efficient than the naive one, as $\mathbf{Q}, \mathbf{K}, \mathbf{V}$ are computed only once from the basis curve, and, more importantly, *a single* $\mathbf{Q}\mathbf{K}^\top \in \mathbb{R}^{N \times N}$ is shared across all curves per head.

### 3.4 VIOLIN ATTENTION

For a single head, we define VIOLIN attention as a decay-masked attention guided by multiple SFCs

$$\mathbf{Y} = \mathrm{Softmax}\left(\alpha \frac{\mathbf{Q}\mathbf{K}^\top}{\sqrt{d}} \odot \mathbf{M}_{\mathrm{VIOLIN}}\right) \mathbf{V},$$

$$\mathbf{M}_{\mathrm{VIOLIN}} = \frac{1}{|\mathcal{C}|} \sum_{c \in \mathcal{C}} \widetilde{\mathbf{M}}_c. \tag{12}$$

Here, $\mathbf{M}_{\mathrm{VIOLIN}}$ is the average of decay masks from all curves $c \in \mathcal{C}$, each first aligned to the basis (Z-curve) order. The matrices $\mathbf{Q}, \mathbf{K}, \mathbf{V}$ are computed from the input $\mathbf{X}$ flattened with respect to the basis. The learnable scalar $\alpha \in \mathbb{R}$ controls how strongly the mask influences attention.

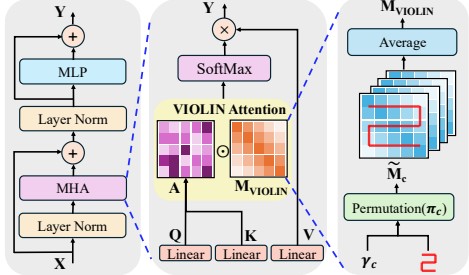

Figure 2: VIOLIN : **(Left)** ViT block with VIOLIN multi-head attention. **(Middle)** Single-head VIOLIN attention. **(Right)** Decay mask $\mathbf{M}_{\mathrm{VIOLIN}}$ formed by averaging masks from curves in $\mathcal{C}$.

For VIOLIN , we use Snake, Zig-zag, Peano, and Hilbert curves together with their transposed variants (Figure 1 **(b2-e2)**) to capture diverse scanning patterns in both row and column major order. This gives the curve set

$$\mathcal{C} = \{\mathrm{Snake}, \mathrm{Zig\text{-}Zag}, \mathrm{Peano}, \mathrm{Hilbert}, \mathrm{Snake}^\top, \mathrm{Zig\text{-}Zag}^\top, \mathrm{Peano}^\top, \mathrm{Hilbert}^\top\}. \tag{13}$$

Each curve $c$ has a decay factor $\gamma_c \in [0, 1]$ for its mask $\mathbf{M}c$, parameterized as $\gamma_c = \mathrm{sigmoid}(\beta_c)$ with learnable $\beta_c \in \mathbb{R}$ for stability, following prior work (Orvieto et al., 2023). n multi-head attention, each head $k$ has its own $\beta_c^k$ and $\alpha^k$, and thus computes $\mathbf{M}c^k$, $\mathbf{M}\mathrm{VIOLIN}{}^k$, and $\pi_c^{-1}(\mathbf{M}_c^k)$ independently. Permutations are applied efficiently via indexing, see code in Appendix G.3. The full VIOLIN block is shown in Figure 2, with further design choices and ablations in Appendix D and Appendix E.

**Parameter and computational overhead** A key advantage of VIOLIN is that it does not introduce significant parameter or computational overhead. As shown in Table 1, the additional cost amounts to only **0.0002% more parameters** and **0.64% more FLOPs** compared to the baseline DeiT-B model with 86M parameters and 55.4G FLOPs. These values are effectively negligible in practice.

Table 1: *Parameter and computational overhead of* VIOLIN *: calculated relative to DeiT-B (86M parameters, 55.4G FLOPs).*

| Metric | Theoretical Computation | % Change (over DeiT-B) |
|---|---|---|
| # Param. | $Lh(|\mathcal{C}|+1)$ | **0.0002%** |
| FLOPs | $\mathcal{O}(LhdN^2)$ | **0.64%** |

Table 2: *GPU memory and inference time comparison:* for DeiT-S and VIOLIN-S at different input resolutions. Batch size is 256.

| Model | GPU Memory (GB) | Runtime (ms/batch) |
|---|---|---|
| DeiT-S ($224 \times 224$) | 0.80 | 206.1 |
| VIOLIN-S ($224 \times 224$) | 0.81 | 233.1 |
| DeiT-S ($512 \times 512$) | 13.88 | 1739.3 |
| VIOLIN-S ($512 \times 512$) | 13.90 | 1789.7 |

To quantify the practical computational cost of VIO-LIN, we report both GPU memory consumption and inference runtime on the same hardware when evaluating a DeiT-S backbone. Measurements are taken for a batch size of 256 at two resolutions: $224 \times 224$ for classification tasks and $512 \times 512$ for dense prediction. As shown in Table 2, VIOLIN closely matches the vanilla DeiT model in both runtime and memory usage. These results are consistent with our theoretical analysis and confirm that VIOLIN introduces only minimal overhead.

## 4 EXPERIMENTS

We evaluate VIOLIN across diverse settings to assess its effect on the spatial awareness of ViTs. Our experiments include fine-tuning on small datasets in Section 4.1, and pretraining small-scale models on ImageNet-1K and on pixel-level CIFAR-100 in Section 4.2. Additional ablations on curve configurations, and decay factors are presented in Section 4.3. Beyond classification, we analyze the strong gains on the Structured group of VTAB and extend evaluation to dense prediction tasks such as detection and segmentation. Overall, VIOLIN consistently improves performance, with the notable benefits in small models and data-scarce regimes.

### 4.1 VTAB-1K FINE-TUNING

The Visual Task Adaptation Benchmark (VTAB) (Zhai et al., 2019) evaluates the adaptability of learned representations to diverse unseen tasks with limited data. It consists of three groups, Natural, Specialized, and Structured, covering 19 datasets from varied domains and semantic categories. In our experiments we use VTAB-1K, a subset with 1,000 examples per task, specifically designed to test model adaptation in data-scarce settings.

We evaluate VIOLIN on small datasets and specialized tasks under two configurations: full fine-tuning and parameter-efficient fine-tuning (PEFT). In both cases, we compare fine-tuning results of the original pretrained models ( Baseline ), and Baseline ⊙ M$_{\text{VIOLIN}}$ where pretrained models are combined with freshly initialized mask before fine-tuning and then optimized jointly with the backbone during fine-tuning. For all models, both baselines and VIOLIN , we use the finetuning implementation from Alkin (2022). For each model, every dataset is first split into a 800/200 train/validation partition to select the optimal learning rate per dataset. We then train on the full dataset using 5 random seeds and report the average of the best 3 runs. The complete set of training hyperparameters is provided in Table 29, and per-dataset results are included in Appendix F.5.

**Full fine-tuning** In the first setting, we test the plug-in capability of VIOLIN by fully fine-tuning pretrained DeiT, DeiT-III, and DINO models across scales ranging from 5M to 630M parameters.

Table 3: *Full fine-tuning results on VTAB-1K*: Comparison of the top-1 accuracies of baseline models and their Baseline ⊙ M$_{\text{VIOLIN}}$ counterparts across the VTAB-1K benchmark. The three task groups are abbreviated as NAT. = Natural, SPE. = Specialized, and STR. = Structured. The values in parentheses ($\cdot$) indicate the accuracy difference compared to the baseline. The best performance within each model pair is highlighted in **bold**. Green highlights the improvement.

| Model | Param. | Top-1 Accuracy (%) | | | | | | | |
|---|---|---|---|---|---|---|---|---|---|
| | | Baseline | | | | Baseline ⊙ M$_{\text{VIOLIN}}$ | | | |
| | | NAT. | SPE. | STR. | Avg. | NAT. | SPE. | STR. | Avg. |
| DeiT-T | 5M | 69.56 | 82.34 | 53.57 | 65.52 | **71.90 (+2.34)** | **83.75 (+1.41)** | **57.50 (+3.93)** | **68.33 (+2.81)** |
| DeiT-S | 22M | 73.64 | 84.30 | 53.44 | 67.38 | **76.06 (+2.42)** | **85.05 (+0.75)** | **58.26 (+4.82)** | **70.46 (+3.08)** |
| DeiT-B | 86M | 76.93 | 85.52 | 57.00 | 70.35 | **77.96 (+1.03)** | **86.29 (+0.77)** | **61.89 (+4.89)** | **72.95 (+2.60)** |
| DeiT-III-S | 22M | 75.13 | 83.63 | 52.92 | 67.57 | **77.03 (+1.90)** | **85.46 (+1.83)** | **61.61 (+8.69)** | **72.31 (+4.74)** |
| DeiT-III-B | 86M | 78.19 | 85.26 | 56.71 | 70.63 | **79.24 (+1.05)** | **86.47 (+1.21)** | **63.03 (+6.32)** | **73.94 (+3.31)** |
| DeiT-III-L | 304M | 88.68 | 84.38 | 51.40 | 67.41 | **90.39 (+1.71)** | **84.68 (+0.30)** | **54.95 (+3.55)** | **69.51 (+2.10)** |
| DeiT-III-H | 632M | 88.15 | 84.18 | 50.70 | 66.91 | **89.10(+0.95)** | **84.43 (+0.25)** | **53.65 (+2.95)** | **68.50 (+1.41)** |
| DINO-S | 22M | 75.35 | 85.09 | 60.65 | 71.21 | **76.26 (+0.91)** | **85.32 (+0.23)** | **61.24 (+0.59)** | **71.84 (+0.63)** |
| DINO-B | 86M | 77.50 | 85.77 | 58.47 | 71.23 | **78.65 (+1.15)** | **86.44 (+0.67)** | **60.84 (+2.37)** | **72.79 (+1.56)** |

During fine-tuning, the VIOLIN decay mask $\mathbf{M}_{\text{VIOLIN}}$ is applied together with the scaling factor $\alpha$ as defined in eq. (12), and the resulting accuracies are reported in Table 3. The freshly initialized mask enables fast adaptation by allowing models to learn task-specific structural biases, which is critical in data-scarce fine-tuning. We also fine-tune the VIOLIN pretrained models from Section 4.2 on the same tasks and noticed that masks learned only during downstream fine-tuning consistently outperform pretrained ones, full results are provided in Appendix F.2.

This property offers a key advantage: VIOLIN *can improve any pretrained model when applied only at fine-tuning.* It removes the need for costly pretraining from scratch and allows model to specialize on the downstream task better. The improvements are substantial, up to **4.7%** on average and **8.7%** on individual group, showing that the spatial bias introduced by VIOLIN enables more effective learning in data-scarce regimes. Moreover, the computational overhead is negligible, and the method generalizes well across training setups, datasets, and model scales, including large models with over 600M parameters.

Table 4: *PEFT results on VTAB-1K with DeiT-B:* # Param. denotes the number of learnable parameters per method. The baseline uses PEFT alone, while VIOLIN combines PEFT with mask fine-tuning.

| Method | # Param. | Avg. Accuracy (%) | |
|---|---|---|---|
| | | Baseline | Baseline $\odot$ $\mathbf{M}_{\text{VIOLIN}}$ |
| Full-FT | 86 M | 70.35 | **72.95 (+2.60)** |
| LoRA | ∼0.3M | 71.04 | **72.55 (+1.41)** |
| DoRA | ∼0.6M | 70.75 | **71.90 (+1.15)** |

**PEFT with VIOLIN** Secondly, we use the PEFT methods LoRA (Hu et al., 2022) and DoRa (Liu et al., 2024a) to fine-tune the DeiT-B model, with results shown in Table 4. In this setting, the VIOLIN mask is freshly initialized and updated alongside the PEFT weights. The additional cost introduced by VIOLIN remains insignificant, only 0.002% additional parameters compared to 0.35% introduced by LoRA. The results demonstrate that VIOLIN can be seamlessly combined with different PEFT methods, further highlighting its applicability and generalizability.

## 4.2 PRETRAINING

**ImageNet-1K pretraining** We pretrain VIOLIN on small-scale models [3] under two paradigms: supervised and self-supervised training, as shown in Table 5. For supervised training, we use DeiT in tiny and small scales, a well established baseline specifically designed for data efficient supervised training. The DeiT paper provides two components: (1) a data-efficient training recipe with tuned augmentations and hyperparameters, and (2) a distillation mechanism that uses a teacher model. In all our DeiT-based pretraining experiments, we use only the training recipe and do not employ any form of distillation. VIOLIN consistently improves performance without any additional tuning, with DeiT-T gaining **0.8%** and DeiT-S achieving a notable **0.9%** improvement, demonstrating strong compatibility. For these models, we adopt Global Average Pooling (GAP) (Lin et al., 2013; Lu et al., 2022) instead of a class token, as GAP is more compatible with VIOLIN , see Appendix E.5 for details.

For self-supervised training, we adopt DINO, a state-of-the-art teacherstudent framework for label-free representation learning, known for its stable training dynamics and strong downstream performance. In our experiments, both teacher and student networks are equipped with VIOLIN attention. In this setup, VIOLIN consistently improves performance across model scales and training durations, yielding gains in both KNN and linear evaluations on ImageNet. For all models, *we strictly follow the original training recipes from the respective papers, without modifying any hyperparameters for* VIOLIN . Baseline accuracies are taken directly from the reported values.

**Ablation studies** In Appendix E, we provide comprehensive ablations on key aspects of VIOLIN attention, all within the same pretraining setup. Appendices E.1 and E.5 examine the effects of global average pooling and positional embeddings, while Appendix E.2 explores different curve configurations, covering all combinations in $\mathcal{C}$, Z-curve only, Manhattan distance-based masking (as used in RMT (Fan et al., 2024)), random curve orderings, and variants without transposed curves. Appendix E.3 compares alternative masking strategies, and Appendix E.4 analyzes key design choices such as initialization, the scaling factor $\alpha$, and fixed vs. learnable decay parameters. Together, these

---

[3]We observed that for ImageNet pretraining with larger models, the performance gains are smaller, which is expected. See Appendix F.1 for numerical results and a detailed explanation.

Table 5: *Pretraining results on ImageNet-1K*: Comparison of the top-1 accuracies of baseline models with their VIOLIN counterparts. The values in parentheses (·) indicate the accuracy difference compared to the baseline. The best performance between each pair of models is highlighted in **bold**. For DINO models, both KNN and linear probe evaluations are reported and (100), (300) indicate the number of training epochs of the models. **(Left)** Supervised training, **(Right)** Self-supervised training. Similar sized CNN baselines are added for comparison.

| Model | # Param. | Top-1 Accuracy (%) Baseline | VIOLIN | Model | | # Param. | Top-1 Accuracy (%) Baseline | VIOLIN |
|---|---|---|---|---|---|---|---|---|
| DeiT-T | 5M | 72.2 | **73.0** (+0.8) | DINO-S (100) | KNN | 22M | 69.3 | **70.0** (+0.7) |
| DeiT-S | 22M | 79.8 | **80.7** (+0.9) | | Linear | | 74.0 | **74.6** (+0.6) |
| ResNet-18 | 12M | 69.8 | | DINO-S (300) | KNN | 22M | 72.8 | **73.4** (+0.6) |
| ResNet-50 | 25M | 76.2 | | | Linear | | 76.1 | **76.4** (+0.3) |

ablations provide a detailed view of each components contribution to the effectiveness of VIOLIN attention. Additionally, in Section F.3, we evaluate the context extrapolation capability of VIOLIN using multi-resolution classification and video generation with a pretrained VIOLIN DINO model, leveraging the natural extrapolation property of the KMS decay mask $M_{VIOLIN}$.

**Pixel-level CIFAR-100 pretraining** Recent work has explored pixel-level tokenization for ViTs (Nguyen et al., 2025; Wang et al., 2025), which provides fine-grained image representations and avoids hand-crafted choices around patch size. However, this setting is challenging because patching is the main source of locality bias in ViTs, removing it makes models more data-hungry and harder to optimize on small or medium sized datasets such as CIFAR-100 (Krizhevsky, 2009). This setting aligns perfectly with the goal of VIOLIN , as it introduces locality into the model independently of the patching process.

Table 6: *Pixel level CIFAR-100 pretraining*: Comparison of the top-1 accuracies of baseline and VIOLIN models.

| Model | # Param. | Avg. Accuracy (%) Baseline | VIOLIN |
|---|---|---|---|
| DeiT-T | 5 M | 60.8 | **68.0** (+7.2) |

On CIFAR-100, when ViT-T is trained using the DeiT ImageNet training recipe, VIOLIN achieves a striking improvement of over **7%** compared to the vanilla pixel-level baseline, as shown in Table 6. This demonstrates that our locality mechanism provides a powerful inductive bias, enabling effective learning in small-data, small-model regimes where standard ViTs collapse. These results highlight both the effectiveness of VIOLIN and the importance of locality awareness for pixel-level ViTs, particularly in resource-constrained scenarios where large-scale pretraining or very long training schedules are impractical.

### 4.3 UNDERSTANDING SPATIAL AWARENESS IN VIOLIN

**Performance gain on the Structured group** The Structured category of VTAB includes tasks that require understanding the spatial structure of the images such as object counting and 3D depth prediction, many of which are derived from simulated environments. These scenes often consist of rendered geometric objects that are simple to humans but differ significantly from images in ImageNet. s a result, success in these tasks often depends on *recognizing positional, orientational, or shape-based information, making local spatial layout especially important.*

As shown in Table 3, the VIOLIN mask provides the largest improvements in this category, with gains of up to **8.69%**, a **16%** relative increase over the baseline. These results highlight the strength of VIOLIN in enhancing spatial capabilities, supporting our claims, and demonstrate its ability to generalize effectively to tasks that depend heavily on spatial structure. In Figure 3, we illustrate images from three datasets in the Structured group with attention heatmaps of DeiT-B models fine-tuned with and without

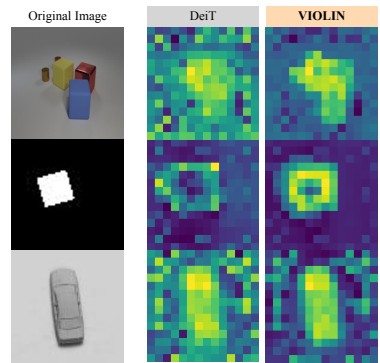

Figure 3: *Attention heatmaps on Structured tasks:* Examples are drawn from three datasets in the Structured group: CLEVR-Count, dSprites-Location, and SmallNORB-Azimuth. All visualizations are taken from layer 12, using the same attention head for each image.

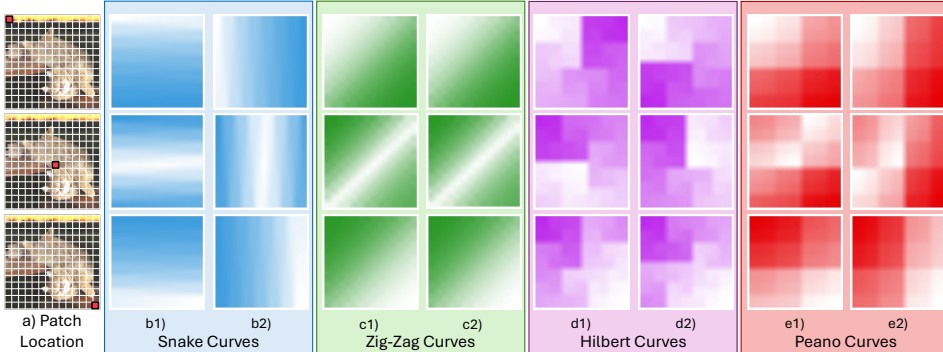

Figure 4: *Mask patterns for different patches:* Visualization of decay mask patterns for three reference patches, top-left, center, and bottom-right, (1st, 2nd and 3rd rows) across all curves and their transposed counterparts. Lighter values indicate stronger spatial relevance, showing more strongly attended regions. **a1)** Reference patch locations, **b1)** Snake, **b2)** Snake$^\top$, **c1)** Zig-zag, **c2)** Zig-zag$^\top$, **d1)** Hilbert, **d2)** Hilbert$^\top$, **e1)** Peano, **e2)** Peano$^\top$ curves.

$\mathbf{M}_{\text{VIOLIN}}$. The comparisons show that models fine-tuned with VIOLIN attend to objects more accurately, suppress noise on irrelevant patches, and produce more uniform responses in background regions, further demonstrating its benefit for spatial understanding. Additional visualizations are provided in Appendix F.4.

**Curve configurations** We examine the individual contribution of each curve by pretraining DeiT-S with all $2^4 = 16$ combinations of four curves (including their transposed variants), with accuracies reported in Table 12. While some combinations yield larger gains, every curve contributes meaningfully, motivating the use of all four in VIOLIN to leverage their complementary spatial information. To illustrate this, Figure 4 visualizes the decay masks for three reference patches (top-left, center, bottom-right) across all curves and their transposes. Lighter regions indicate stronger attention, and the distinct patterns show how different curves bias the model toward diverse spatial regions.

We further analyze the learned decay parameters $\gamma_c$ for DeiT-B in Figure 7, observing that most remain close to one, indicating active use of long-range spatial information. Smaller values act as implicit curve selection, as these decay masks would contribute to the average minimally, with certain layers and heads emphasizing particular curves. Finally, additional attention heatmaps and visualizations of sequences flattened by different curves are provided in Appendix F.4.

**Comparison against other inductive bias methods** In Table 7, we provide an extended comparison of various locality-enforcing baselines on the Structured group in the fine-tuning setting. For each approach, we use the same pretrained DeiT-B backbone and initialize the corresponding locality mechanism on top of it, ensuring that all models start from an identical initialization. All methods are then fine-tuned under the same protocol, using the hyperparameter set described in Table 29.

Table 7: *Comparison of locality methods:* The pretrained DeiT-B model fine-tuned with different locality methods on the VTAB Structured group. Best result is highlighted on **bold**.

| Method | # Extra Parameters | Structured Avg. (%) |
|---|---|---|
| Baseline (DeiT-B) | – | 57.00 |
| VIOLIN | ~1.3K | **61.89** |
| Additive $\mathbf{M}_{\text{VIOLIN}}$ | ~1.3K | 61.34 |
| Swin RPB | ~105K | 61.58 |
| i-RPE QKV | ~115K | 61.45 |
| LocalViT | ~6.2M | 61.50 |
| Manhattan Mask | ~0.4K | 58.37 |
| Single SFC ($\mathbf{M}_{\text{Peano}}$) | ~0.4K | 61.63 |
| Random Curve ($\mathbf{M}_{\text{Random}}$) | ~0.4K | 61.43 |

These results show that while most locality priors offer some improvement, VIO-LIN achieves the strongest gains with minimal overhead. This indicates that the improvements come specifically from the usage of multiple SFC curves, rather than from the presence of any local bias. Moreover, the results highlight VIOLIN's effectiveness as a plug-and-play spatial prior in small-data finetuning regimes. Full implementation details, initialization choices, and per-dataset results are provided in Appendix F.6.

Table 8: *Results on dense prediction tasks:* (**Left**) mIoU scores on semantic segmentation on ADE20K with DeiT-B model. (**Right**) box AP and mask AP scores on object detection and instance segmentation on COCO with Swin-T.

| Backbone | mIoU | | Backbone | Baseline | | **Baseline** $\odot$ $\mathbf{M_{VIOLIN}}$ | |
| | Baseline | **Baseline** $\odot$ $\mathbf{M_{VIOLIN}}$ | | box AP | mask AP | box AP | mask AP |
| --- | --- | --- | --- | --- | --- | --- | --- |
| DeiT-B | 45.24 | **45.80 (+0.56)** | Swin-T | 42.7 | 39.3 | **42.8 (+0.1)** | **39.7 (+0.4)** |

**Dense prediction tasks** To assess the capabilities of VIOLIN beyond classification, we evaluate it on semantic segmentation and object detection. For both tasks, baseline and VIOLIN enhanced models are trained under identical setups to ensure fair comparison, with results reported in Table 8. These experiments also highlight the flexibility of $\mathbf{M_{VIOLIN}}$, which naturally generalizes to arbitrary input shapes, enabling resolution expansion and non-square images.

For semantic segmentation, we use ADE20K (Zhou et al., 2017; 2019), a challenging scene parsing dataset, implemented in the `mmsegmentation` framework (Contributors, 2020). The backbone is an ImageNet pretrained DeiT-B model combined with UPerNet (Xiao et al., 2018). The $\mathbf{M_{VIOLIN}}$ mask is freshly initialized at fine-tuning, and training is performed for 80k iterations with batch size 16. As reported in Table 8, VIOLIN achieves a **+0.56** mIoU improvement, further demonstrating that spatial priors help ViTs adapt effectively to dense prediction tasks.

For object detection, we experiment on COCO (Lin et al., 2015) using the `mmdetection` framework (Chen et al., 2019). The backbone is an ImageNet pretrained Swin-T (Liu et al., 2021), paired with Mask R-CNN (He et al., 2017) as the detector. As in segmentation, the VIOLIN mask $\mathbf{M_{VIOLIN}}$ is freshly initialized at fine-tuning, and models are trained with a $1\times$ schedule and batch size 16. As shown in Table 8, VIOLIN improves performance by **+0.4** mAP over the baseline, showing that spatial priors from space-filling curves enhance object localization.

## 5 CONCLUSION AND FUTURE DIRECTIONS

In this work, we introduced VIOLIN, a masked attention mechanism inspired by the decay masks of Linear Transformers and the perspective of flattening via space filling curves. By integrating diverse spatial patterns into a unified decay mask, VIOLIN enhances the understanding of relative spatial relationships without altering the training recipe, or introducing a significant computational cost.

Our experiments show that VIOLIN is particularly effective in small models and data-scarce settings, where spatial inductive bias is most critical. It also serves as a plug-and-play module that can be applied only during fine-tuning, combining seamlessly with parameter-efficient methods. More broadly, VIOLIN emphasizes the overlooked role of patch ordering and spatial priors in ViT design, offering a lightweight and practical approach to strengthen locality in vision transformers.

**Future directions** Since VIOLIN operates directly on the attention scores, it can be used in any setting where spatial relationships are important and a global attention mechanism is used. This opens up many exciting future directions, including applications to depth estimation, super-resolution, tracking, and even video understanding. VIOLIN also opens several promising directions, such as dynamic or task-adaptive curve selection, as well as to domains such as video, multimodal learning, and data-scarce applications like medical imaging or satellite analysis. These settings offer promising opportunities to further explore the impact of explicit spatial priors in vision backbones.

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

# Appendix

## Table of Contents

# A  NOTATIONS

In Table 9, we summarize the notations used in the paper.

Table 9: *Notations*: Summary of notations used throughout the paper.

| Definition | Notation |
|---|---|
| Image | $\mathcal{I} \in \mathbb{R}^{H \times W \times d}$ |
| Curves set | $\mathcal{C}$ |
| Curve ID | $c \in \mathcal{C}$ |
| Flattening operator with curve $c$ | $F_c(\mathcal{I}) : \mathbb{R}^{H \times W} \to \mathbb{R}^N$ |
| Flattened image with curve $c$ | $\mathbf{X}_c \in \mathbb{R}^{N \times d}$ |
| Permutation from curve $c_1$ to $c_2$ | $\pi_{c_1 \to c_2}(i)$ |
| Permutation matrix from curve $c_1$ to $c_2$ | $\mathbf{P}_{c_1 \to c_2} \in \mathbb{R}^{N \times N}$ |
| Decay mask for basis curve (Z-curve) | $\mathbf{M} \in \mathbb{R}^{N \times N}$ |
| Decay mask for curve $c$ | $\mathbf{M}_c \in \mathbb{R}^{N \times N}$ |
| Permuted decay mask for curve $c$ | $\widetilde{\mathbf{M}_c} \in \mathbb{R}^{N \times N}$ |
| Average of all decay masks for all curves | $\mathbf{M}_{\text{VIOLIN}} \in \mathbb{R}^{N \times N}$ |
| Average mask scaling parameter | $\alpha \in \mathbb{R}$ |
| Decay parameter for mask $\mathbf{M}_c$ | $\gamma_c \in \mathbb{R}$ |
| Queries, keys, values | $\mathbf{Q}, \mathbf{K}, \mathbf{V} \in \mathbb{R}^{N \times d}$ |

# B  EXTENDED BACKGROUND

## B.1  VITS AND SPATIAL PRIORS

ViTs are powerful alternatives to Convolutional Neural Networks (CNNs) (O'Shea & Nash, 2015), but their design comes with a fundamental limitation: a lack of inherent spatial inductive bias. Unlike CNNs, where convolutions naturally encode locality and translation equivariance, ViTs treat images as sequences of independent patches. Spatial relations must therefore be inferred entirely from data, with positional embeddings and patching serving as the primary source of spatial information (Dosovitskiy et al., 2021; Yuan et al., 2021). This design provides ViTs with flexibility in modeling global dependencies, however it also removes the strong inductive priors that are especially critical in data-scarce settings (d'Ascoli et al., 2021; Wu et al., 2021b).

The absence of spatial inductive bias makes ViTs particularly fragile and data hungry when model capacity or training data is limited. Small ViTs trained on large datasets often underperform compared to CNNs, since they cannot rely on built-in locality to efficiently capture low-level spatial features (Touvron et al., 2021; Yuan et al., 2021). In contrast, when both models and datasets are sufficiently large, and training is long enough, ViTs can learn these biases directly from data. For instance, large-scale training on ImageNet-21k (Ridnik et al., 2021) or JFT (Sun et al., 2017) demonstrates that ViTs can eventually match or surpass CNNs, but this comes at considerable computational and data cost (Dosovitskiy et al., 2021; Touvron et al., 2021). Therefore, spatial inductive bias is highly beneficial in practice, especially for downstream tasks, resource-constrained scenarios and small scale models.

Motivated by this tradeoff, various approaches have emerged to reintroduce spatial priors into transformer architectures. Hierarchical models such as Swin Transformer (Liu et al., 2021; 2022) and Pyramid Vision Transformer (PVT) (Wang et al., 2021; 2022b) adopt CNN-like multi-scale processing, enabling more efficient capture of local and global dependencies. Similarly, T2T-ViT (Yuan et al., 2021) progressively aggregates tokens to embed local structure. These designs restore the inductive biases of locality and scale, improving performance in regimes where pure ViTs struggle.

Another line of work incorporates convolutions directly into the transformer pipeline. Convolutional hybrids such as CvT (Wu et al., 2021b), ConViT (d'Ascoli et al., 2021), and CMT (Guo et al., 2022) explicitly embed local connectivity into the attention mechanism or token embedding process, bridging the gap between CNNs and ViTs. Other methods explore novel locality-aware mechanisms, including vicinity attention (Zhang et al., 2023), shuffle-based spatial mixing (Huang et al., 2021),

and localized attention modules (Li et al., 2021; Chu et al., 2021). Even more recent innovations, such as RMT (Fan et al., 2024), propose decay masks inspired by RetNet (Sun et al., 2023) to enforce local inductive constraints.

Despite their effectiveness, most of these approaches achieve improved spatial priors by directly modifying the ViT architecture such as embedding convolutions into tokenization, or restructuring the model into hierarchical stages. While such changes enhance locality, they also increase design complexity, reduce modularity, and often require pretraining from scratch on large datasets to fully realize their benefits. This makes them less practical in settings where one wishes to reuse widely available pretrained vanilla ViTs. In contrast, methods that can inject spatial inductive bias without altering the base architecture, for instance, during fine-tuning, offer a more lightweight and flexible alternative, enabling broader applicability to downstream tasks and smaller models without sacrificing compatibility with existing pretrained checkpoints.

What remains missing is a simple mechanism to bridge this gap: an approach that can utilize already trained ViTs while still strengthening their spatial priors, which can be achieved via VIOLIN with close to zero additional cost.

## B.2 LINEAR TRANSFORMERS

Linear attention is mathematically equivalent to an RNN (Katharopoulos et al., 2020)

$$\mathbf{S}_i = \mathbf{S}_{i-1} + \mathbf{k}_i^\top \mathbf{v}_i, \quad \mathbf{y}_i = \mathbf{q}_i^\top \mathbf{S}_i \quad \Leftrightarrow \quad \mathbf{Y} = (\mathbf{Q}\mathbf{K}^\top \odot \mathbf{L}_{\text{Causal}})\mathbf{V}, \tag{14}$$

where $\mathbf{S}_i \in \mathbb{R}^{d \times d}$ represents the hidden state of the Linear Transformer in its equivalent RNN form and $\mathbf{L}_{\text{Causal}} \in \mathbb{R}^{N \times N}$ is lower triangular matrix of ones.

Building on that, Linear Transformers with a scalar decay factor commonly take the following recurrent form:

$$\mathbf{S}_i = \boldsymbol{\Lambda}_i \mathbf{S}_{i-1} + \mathbf{k}_i^\top \mathbf{v}_i, \quad \mathbf{u}_i = \mathbf{q}_i^\top \mathbf{S}_i \tag{15}$$

with hidden state $\mathbf{S}_i$ and output $\mathbf{y}_i$. Here, the behavior of the model is determined by the choice of the decay parameter $\boldsymbol{\Lambda}_i$. It is also standard practice to apply a non-linearity to the queries and keys, such that $\mathbf{Q}, \mathbf{K} = \phi(\mathbf{W}_Q\mathbf{X}), \phi(\mathbf{W}_K\mathbf{X})$, and to scale attention in relation to past tokens, as discussed in Katharopoulos et al. (2020).

**No decay** In vanilla Linear Transformers (eq. (2)), there is no decay term, or equivalently $\boldsymbol{\Lambda}_i = \mathbf{I}$ where $\mathbf{I}$ is the identity matrix. As a result, these models do not encode relative positional information. Performer (Choromanski et al., 2021) is a representative example, using Random Fourier Features (RFF) (Peng et al., 2021) as the non-linear function $\phi(\cdot)$, without any form of decay mechanism.

**Non input-dependent decay** A key example in this category is RetNet (Sun et al., 2023), which employs a fixed scalar decay parameter $\boldsymbol{\Lambda}_i = \gamma$. This introduces a locality bias in the attention computation, but the decay remains constant and independent of the input sequence.

**Input-dependent decay** Several recent linear transformers in the NLP domain fall into this category, where the decay parameter $\boldsymbol{\Lambda}_i = g(\mathbf{x}_i)$ is a function of the input and thus varies across tokens. For example, DeltaNet (Yang et al., 2024) defines the decay using the Delta Rule (Schlag et al., 2021) as $\boldsymbol{\Lambda}_i = \mathbf{I} - \mathbf{k}_i\mathbf{k}_i^\top$, while Gated RFA (Peng et al., 2021) uses an input-dependent scalar decay of the form $\boldsymbol{\Lambda}_i = \sigma(\mathbf{W}\mathbf{x}_i)$, where $\sigma(\cdot)$ is the sigmoid function and $\mathbf{W} \in \mathbb{R}^d$, resulting in a scalar decay value per token.

**Selective SMMs** This category of models is closely related to linear transformers with input-dependent decay. A prominent example is Mamba (Gu & Dao, 2024), which can be interpreted as a linear transformer with an input-dependent diagonal matrix as the decay parameter $\boldsymbol{\Lambda}_i$ (Yang et al., 2023). Mamba-2 (Dao & Gu, 2024), a simplified variant, further refines this by using an exponential formulation for the decay factor: $\boldsymbol{\Lambda}_i = \exp(-\exp(\mathbf{W}\mathbf{x}_i))$, enabling a more stable and expressive modeling of token-wise recurrence.

### B.3 SPACE FILLING CURVES

SFCs have diverse applications across various domains, including image compression and generation (Wang et al., 2022a; Dafner et al., 2000), point cloud processing (Chen et al., 2023), data mining (Bhm, 2020), and data movement (Walker & Skjellum, 2023). In this section, we define the curves used in this study as flattening operation $F_c$ for each curve. The definitions are adapted from (Sagan, 1994; Peano, 1990; Hilbert, 1935; Zhao et al., 2024).

**Z-curve**  The Z-curve, also known as sweep, row-major order, or raster scan, is the simplest and most widely used method for flattening a 2D image into a 1D sequence. It scans the image row by row, from top to bottom and left to right within each row. More concretely, for an image with width $W$, the flattening function can be defined as

$$F_z(i, j) = iW + j. \tag{16}$$

This flattening order is the default scanning method in many vision models, including ViTs. As a result, we use it as our basis in the paper.

**Snake Curve**  The snake curve, also known as boustrophedon order (Fernau et al., 2015), is a variation of the Z-curve that alternates the scanning direction across rows. Even-indexed rows are traversed left to right, while odd-indexed rows are traversed right to left, creating a continuous snake path through the image. The flattening function is given by:

$$F_{\text{snake}}(i, j) = \begin{cases} i \cdot W + j & \text{if } i \bmod 2 = 0 \\ i \cdot W + (W - 1 - j) & \text{if } i \bmod 2 = 1 \end{cases} \tag{17}$$

This curve has a simplicity similar to the Z-curve while reducing long jumps between the end of one row and the beginning of the next. It is utilized in various applications, including image processing and path planning, due to its efficiency in covering areas without unnecessary repositioning.

**Zig-zag Curve**  The Zig-zag curve (Wallace, 1992) is a diagonal scanning pattern that visits patches of an image along consecutive diagonals, alternating direction at each level. More concretely, with an image of size $H \times W$, for each diagonal $g \in \{0, \ldots, H + W - 2\}$, it scans the elements where $i + j = g$, from top-right to bottom-left on odd-numbered diagonals and from bottom-left to top-right on even-numbered ones. In other words, for each diagonal $g$, let the set of valid coordinates on that diagonal be $D_g = \{(i, j) \mid i + j = g, \ 0 \le i < H, \ 0 \le j < W\}$. Then the ordering of $F_{\text{zigzag}}(i, j)$ can be defined by

$$F_{\text{zigzag}}(i, j) = \left( \sum_{k=0}^{g-1} |D_k| \right) + \text{offset}_g(i, j), \tag{18}$$

where $|D_k|$ is the length of the diagonal and $\text{offset}_g(i, j)$ is

$$\text{offset}_g(i, j) = \begin{cases} \#\{(i', j') \in D_g \mid j' < j\} & \text{if } g \bmod 2 = 0, \\ \#\{(i', j') \in D_g \mid j' > j\} & \text{if } g \bmod 2 = 1. \end{cases}$$

The zig-zag curve is most commonly used in applications where frequency components are spatially grouped such as the JPEG compression standard to serialize the block of discrete cosine transform (DCT) coefficients, to ensure that low-frequency components that carry the most information appear early in the sequence.

**Hilbert Curve**  The Hilbert curve (Hilbert, 1935) recursively divides the space into quadrants and connects them in a continuous path that fills the entire 2D grid. Similar to Peano curve, the Hilbert curve is most naturally defined on square images of size $2^p \times 2^p$ where the recursive quadrant-based construction aligns with the binary structure of the coordinates. The flattening function $F_{\text{hilbert}}(i, j)$ does not have a simple closed-form expression, but can be computed via recursive or bitwise algorithms, for example, Butz or Moore methods (Butz, 1969; Moore, 1900).

For an image of size $H \times W$ with $H = W = 2^p$, we can define the Hilbert curve flattening function as

$$F_{\text{hilbert}}(i, j) = \sum_{k=1}^{n} q_k \cdot 4^{n-k} \tag{19}$$

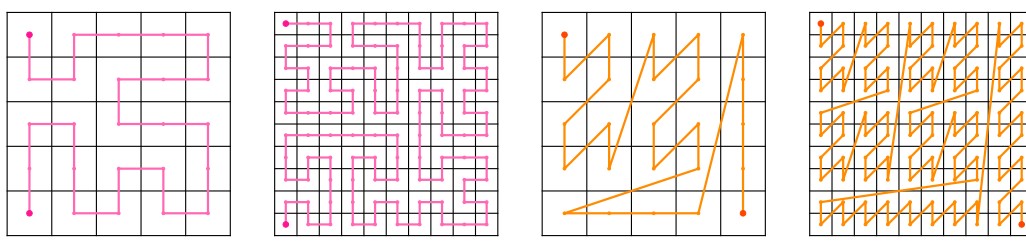

(a) Hilbert on $5 \times 5$ grid.  (b) Hilbert on $10 \times 10$ grid.  (c) Peano on $5 \times 5$ grid.  (d) Peano on $10 \times 10$ grid.

Figure 5: *Extension of Hilbert and Peano curves:* Visualization of how Hilbert and Peano curves extend to non-power-of-2 grids.

where $q_1 q_2 \cdots q_n$ is the base-4 Hilbert index corresponding to the normalized pixel center:

$$\left( \frac{i}{2^n} + \frac{1}{2^{n+1}}, \quad \frac{j}{2^n} + \frac{1}{2^{n+1}} \right) \in [0,1)^2 \tag{20}$$

Each digit $q_k \in \{0, 1, 2, 3\}$ represents the quadrant at level $k$ in the recursive Hilbert construction.

Points that are close in 2D space tend to remain close in 1D, which makes it especially valuable in image processing, spatial indexing, and contexts where locality is significant.

**Peano Curve**   The Peano curve, also called Z-order curve or Morton curve, (Peano, 1990) is a recursive scanning approach that preserves spatial locality by interleaving the binary representations of the row and column indices. It is particularly well-suited to square grids of size $2^p \times 2^p$ as the bit structure of the coordinates aligns naturally with the recursive subdivisions of the curve.

For $H = W = 2^p$, let $(i, j) \in \{0, \ldots, 2^p - 1\}^2$ be the pixel coordinates, and we can write their binary expansions:

$$i = \sum_{k=0}^{n-1} i_k \cdot 2^k, \quad j = \sum_{k=0}^{n-1} j_k \cdot 2^k \quad \text{with } i_k, j_k \in \{0, 1\} \tag{21}$$

$$F_{\text{peano}}(i, j) = \text{interleave\_bits}(i, j) = \sum_{k=0}^{p-1} \left( j_k \cdot 2^{2k+1} + i_k \cdot 2^{2k} \right) \tag{22}$$

As it can be constructed bitwise, it is computationally efficient and commonly used in applications like image tiling, spatial databases, and quadtree indexing.

**Remark:**   While the Peano and Hilbert curves are most naturally defined on square grids with power-of-two dimensions, they can be easily extended to arbitrary image sizes by truncating higher-order bits, using padding, clipping, or floating-point mapping techniques (Cerveny, 2025; Sasidharan et al., 2015). In Figure 5, we visually show how to extend these curves to non-power-of-2 cases with codes provided in Appendix G.3.

**Flattening with transposed curves**   Standard SFCs are typically defined over fixed scans using row-major or column-major orderings. To increase the diversity of locality preserving patterns without incurring additional cost, we introduce transposed variants of standard SFCssuch as column-major Snake or vertical Zig-Zag. These variants simply swap coordinates during traversal. We define the flattened image under a transposed curve as:

$$\mathbf{X}_{c^\top}[n] = \mathcal{I}[i, j] \quad \text{where} \quad n = F_{c^\top}(i, j) = F_c(j, i). \tag{23}$$

Accordingly, we expand our curve set to include these rotated versions, resulting in the final VIOLIN curve set:

$$\mathcal{C} = \{\text{Snake, Zig-Zag, Peano, Hilbert, Snake}^\top, \text{Zig-Zag}^\top, \text{Peano}^\top, \text{Hilbert}^\top\} \tag{24}$$

### B.4 Locality via decay mask

**Decay mask structure**  An example of a $4 \times 4$ causal decay mask with non-input-dependent decay factor, as used in RetNet (Sun et al., 2023), is

$$\mathbf{M}_{\text{Causal}} = \begin{bmatrix} 1 & & & \\ \gamma & 1 & & \\ \gamma^2 & \gamma & 1 & \\ \gamma^3 & \gamma^2 & \gamma & 1 \end{bmatrix}, \qquad \mathbf{M}_{\text{Causal}}[i,j] = \begin{cases} \gamma^{i-j} & i \geq j \\ 0 & i < j \end{cases} \tag{25}$$

As seen in the causal decay mask above, the decay masking the attention $\mathbf{M}_{\text{Causal}}[i,j]$ depends only on the difference between $i$ and $j$, specifically $\mathbf{M}_{\text{Causal}}[i,j] = \gamma^{|i-j|}$. which reflects the locality information in the causal decay mask.

As an extension for bidirectional tasks, such as image classification, the causal mask can be extended to a full Toeplitz decay mask, as shown in (Afzal et al., 2025):

$$\mathbf{M} = \begin{bmatrix} 1 & \gamma & \gamma^2 & \gamma^3 \\ \gamma & 1 & \gamma & \gamma^2 \\ \gamma^2 & \gamma & 1 & \gamma \\ \gamma^3 & \gamma^2 & \gamma & 1 \end{bmatrix}, \qquad \mathbf{M}[i,j] = \gamma^{|i-j|} \tag{26}$$

in this case, the attention between each pair of tokens $i$ and $j$ is masked based on their distance $|i - j|$. Additionally, the decay factor $0 < \gamma < 1$ is bounded between to ensure that $\mathbf{M}[i,j]$ does not overflow and remains stable (Orvieto et al., 2023).

**Extrapolation capabilities of decay mask**  The decay mask $\mathbf{M}$ can easily be extrapolated beyond the context length (Dao & Gu, 2024; Sun et al., 2023) because $\mathbf{M}[i,j] = \gamma^{|i-j|}$ is independent of the sequence length. This is especially useful since we can change the resolution of images during inference without needing to interpolate or extrapolate the position embeddings (Dosovitskiy et al., 2021; Caron et al., 2021). This capability is particularly valuable when generating videos for object tracking in VIOLIN DINO.

### B.5 Efficiency of Toeplitz decay mask

As mentioned in the background Appendix B.2, the decay parameter $\gamma$ can be input dependent as well, which means that it is extracted for each token as:

$$\gamma_i = g(\mathbf{W}_\gamma \mathbf{x}_i), \quad \mathbf{M}[i,j] = \gamma_j \gamma_{j+1} ... \gamma_i = \prod_{k=j}^{i} \gamma_k \tag{27}$$

with $g(.)$ being a bounded function such that $0 < g(x) < 1$ (i.e. sigmoid). This results in each element of the decay mask $\mathbf{M}[i,j]$ representing the cumulative product of decay contributions from all tokens between positions $i$ and $j$ leading to input-dependent decay masks. While these type of masks can offer finer-grained control, they are slower to train, requiring $\mathcal{O}(\log(N))$ time points to compute (Gu & Dao, 2024; Dao & Gu, 2024), consume more memory, and must be dynamically constructed during inference. In contrast, input-independent decay masks such as the one used in VIOLIN are much more efficient. We adopt the decay mask in VIOLIN as it is faster to train, memory-efficient (requiring only a single learned scalar $\gamma$ per curve), and eliminates the need for recomputation during inference. This simple scalar-based design still performs effectively and achieves strong results in practice (Afzal et al., 2025).

### B.6 Connections of Violin to other models

As VIOLIN is inspired by the forget gate (also known as the decay mask) in Linear Transformers, it shares strong connections with these models and their adaptations for vision tasks. Below, we highlight some of the most relevant connections:

**RMT**  RMT (Fan et al., 2024) also introduces a decay mask (via Manhattan distance) to enhance the spatial awareness of ViTs, addressing a similar challenge. However, it differs from VIOLIN in key

ways. RMT uses only a single flattening strategy and applies a fixed distance metric (Manhattan), while VIOLIN generates multiple masks based on different SFCs and defines a KacMurdockSzeg (KMS) matrix for the decay. Architecturally, VIOLIN is a modular attention mechanism that can be plugged into various ViT backbones, whereas RMT is a standalone model. We also conducted an ablation using the Manhattan distance decay as in RMT, and found it underperforms compared to VIOLIN . Detailed results are provided in Table 13.

**FoX** FoX, or Forgetting Transformer (Lin et al., 2025), is designed for causal sequence modeling, specifically to capture long-range dependencies in the NLP domain. It uses an input-dependent causal decay mask, as shown in eq. (27), which differs significantly from VIOLIN in both application domain and mask design. Moreover, the perspective central to VIOLIN , based on flattening and scanning via space-filling curves, does not appear in FoX, as it operates in the NLP setting rather than vision tasks.

**Vision Linear Transformer** This class includes models such as Vision LSTM (Alkin et al., 2024), Vision Mamba (Zhu et al., 2024), and VMamba (Liu et al., 2024b), which are related to VIOLIN due to their use of different scanning strategies primarily based on the Z-curve in both standard and transposed (horizontal and vertical) directions. However, these models significantly differ from VIOLIN in architecture, as they are based on SSMs like Mamba (Gu & Dao, 2024) or other linear attention mechanisms, rather than softmax-based Transformers. In contrast, VIOLIN is a softmax-based masked attention module that can be easily integrated into various ViT backbones. In this study, we apply VIOLIN to DeiT, DeiT-III, and DINO as representative examples.

**MAE** Masked Auto Encoders (MAE) (He et al., 2022) apply random input masking as a pretraining objective, dropping patches and training the model to reconstruct them. This masking affects only the input and does not influence attention computation. In contrast, VIOLIN applies structured masking within the attention mechanism, using decay masks based on space-filling curves to rescale attention scores, without dropping tokens or reconstructing inputs. It serves as a spatial inductive bias, guiding the model to attend more to nearby regions without altering the input or training objective.

# C PROOFS

## C.1 ATTENTION IS PERMUTATION EQUIVARIANT

**Claim C.1.** Attention without positional embeddings is permutation-equivariant. That is,

$$A(\pi(\mathbf{X})) = \pi(A(\mathbf{X})) \tag{28}$$

where $A(\cdot)$ is the output of the attention mechanism, and $\pi(\cdot)$ denotes a permutation of the sequence.

*Proof.* Let $\mathbf{X} \in \mathbb{R}^{N \times d}$ be the input sequence with $N$ tokens and model dimension $d$. The attention is defined as

$$\mathbf{Q} = \mathbf{X}\mathbf{W}_Q, \quad \mathbf{K} = \mathbf{X}\mathbf{W}_K, \quad \mathbf{V} = \mathbf{X}\mathbf{W}_V, \qquad A(\mathbf{X}) = \text{Softmax}\left(\frac{\mathbf{Q}\mathbf{K}^\top}{\sqrt{d}}\right)\mathbf{V}. \tag{29}$$

Let $\pi$ be a permutation of the input sequence, represented by a permutation matrix $\mathbf{P} \in \mathbb{R}^{N \times N}$ such that $\pi(\mathbf{X}) = \mathbf{P}\mathbf{X}$ and $\mathbf{P}\mathbf{P}^\top = \mathbf{I}$. Then

$$\pi(\mathbf{Q}) = \mathbf{P}\mathbf{X}\mathbf{W}_Q = \mathbf{P}\mathbf{Q}, \quad \pi(\mathbf{K}) = \mathbf{P}\mathbf{K}, \quad \pi(\mathbf{V}) = \mathbf{P}\mathbf{V}. \tag{30}$$

Now compute the attention on the permuted input

$$A(\pi(\mathbf{X})) = \text{Softmax}\left(\frac{(\mathbf{P}\mathbf{Q})(\mathbf{P}\mathbf{K})^\top}{\sqrt{d}}\right)(\mathbf{P}\mathbf{V}) = \text{Softmax}\left(\frac{\mathbf{P}\mathbf{Q}\mathbf{K}^\top\mathbf{P}^\top}{\sqrt{d}}\right)\mathbf{P}\mathbf{V} \tag{31}$$

Since softmax is applied row-wise and permutation matrices preserve row-wise operations, we can factor $\mathbf{P}$ out

$$A(\pi(\mathbf{X})) = \mathbf{P}\,\text{Softmax}\left(\tfrac{\mathbf{Q}\mathbf{K}^\top}{\sqrt{d}}\right)\underbrace{\mathbf{P}^\top\mathbf{P}}_{\mathbf{I}}\mathbf{V} = \mathbf{P}\,\text{Softmax}\left(\tfrac{\mathbf{Q}\mathbf{K}^\top}{\sqrt{d}}\right)\mathbf{V} = \mathbf{P}A(\mathbf{X}) = \pi(A(\mathbf{X})) \tag{32}$$

Thus, attention is permutation-equivariant in the absence of positional embeddings. $\qquad\square$

## C.2   SFCs IN DECAY MASK ARE A DISTANCE METRIC

**Claim C.2.** Let $\mathbf{X}_{c_1} \in \mathbb{R}^{N \times d}$ be the flattened image using a space-filling curve $c_1$, with the sequence indexed by $i, j, k \in \{0, \ldots, N - 1\}$. Any permutation $\pi_{c_2}$, corresponding to a new flattening order defined by a different curve $c_2$, when applied to $\mathbf{X}_{c_1}$, induces a new sequence order. In this new order, the term $|\pi(i) - \pi(j)|$ satisfies the non-negativity, identity of indiscernibles, symmetry and triangle inequality properties of a distance metric between tokens $i$ and $j$.

*Proof.* To show that $|\pi(i) - \pi(j)|$ is a valid distance metric, we verify that it satisfies the standard properties of a metric:

*Non-negativity:* For all $i, j$, we have

$$|\pi(i) - \pi(j)| \geq 0 \tag{33}$$

since absolute values are always non-negative.

*Identity of indiscernibles:*

$$|\pi(i) - \pi(j)| = 0 \iff \pi(i) = \pi(j) \iff i = j \tag{34}$$

because $\pi$ is a permutation (i.e., a bijective function), so $\pi(i) = \pi(j)$ implies $i = j$.

*Symmetry:*

$$|\pi(i) - \pi(j)| = |\pi(j) - \pi(i)| \tag{35}$$

by the symmetry of absolute value.

*Triangle inequality:* For any $i, j, k \in \{0, \ldots, N - 1\}$,

$$|\pi(i) - \pi(j)| \leq |\pi(i) - \pi(k)| + |\pi(k) - \pi(j)| \tag{36}$$

holds due to the triangle inequality property of absolute values.

Therefore, $|\pi(i) - \pi(j)|$ satisfies all the conditions of a distance metric. This property is particularly interesting because the term $|\pi(i) - \pi(j)|$ appears as the exponent in the decay mask, leading to $\mathbf{M}_{c_2}[i, j] = \gamma^{|\pi(i) - \pi(j)|}$. As a result, taking the logarithm of the decay mask yields a distance matrix, $\log(\mathbf{M}_{c_2}[i, j]) = |\pi(i) - \pi(j)| \cdot \log(\gamma)$ thus, $\log(\mathbf{M}_{c_2})$ is a scaled distance matrix, encoding relative positional distances under the permutation induced by curve $c_2$.   $\square$

## C.3   VIOLIN SFC FLATTENING ONLY REFLECTS IN DECAY MASK

**Claim C.3.** Let the input sequence flattened using a base space-filling curve (e.g., Z-curve) be denoted by $\mathbf{X} \in \mathbb{R}^{N \times d}$, and let the output of VIOLIN attention be $\mathbf{Y} \in \mathbb{R}^{N \times d}$, computed as:

$$\mathbf{Y} = \text{Softmax}\left(\alpha \frac{\mathbf{Q}\mathbf{K}^\top}{\sqrt{d}} \odot \mathbf{M}\right) \mathbf{V} \tag{37}$$

where $\mathbf{M} \in \mathbb{R}^{N \times N}$ is the base decay mask with entries $\mathbf{M}[i, j] = \gamma^{|i-j|}$.

Now, let $\mathbf{X}_c = \pi_c(\mathbf{X})$ be the input sequence reordered using a space-filling curve $c$, with permutation $\pi_c$. Then, the output of the VIOLIN attention for the permuted input $\mathbf{X}_c$, re-ordered back to the original (basis) input order, is given by:

$$\widetilde{\mathbf{Y}} = \text{Softmax}\left(\alpha \frac{\mathbf{Q}\mathbf{K}^\top}{\sqrt{d}} \odot \pi_c(\mathbf{M})\right) \mathbf{V} \tag{38}$$

where $\pi_c(\mathbf{M}) = \mathbf{M}[\pi_c(i), \pi_c(j)]$ denotes the decay mask permuted along both rows and columns according to the curve $c$.

*Proof.* It is easy to see that flattening the input $\mathcal{I}$ into a sequence $\mathbf{X}_{c_1}$ using any space-filling curve $c_1$ defines a one-to-one mapping from the 2D grid to a 1D sequence. Therefore, there exists a permutation $\pi_{c_1 \to c_2}$ and an associated permutation matrix $\mathbf{P}_{c_1 \to c_2}$ such that the sequence obtained by flattening with another curve $c_2$ is given by:

$$\mathbf{X}_{c_2} = \mathbf{P}_{c_1 \to c_2} \mathbf{X}_{c_1} \tag{39}$$

Now, considering $c_1$ as the z-Curve (our basis flattening), and renaming $c_2$ simply as $c$, we simplify the notation as follows:

$$\pi_{c_1 \to c_2} = \pi_c, \quad \mathbf{P}_{c_1 \to c_2} = \mathbf{P}_c, \quad \mathbf{X}_c = \pi_c(\mathbf{X}) = \mathbf{P}_c\mathbf{X} \tag{40}$$

From eq. (30) we know that permuting the input $\mathbf{X}$ will result in permutation of query, key and value matrices so for the input $\mathbf{X}_c$ the attention presented at eq. (37) is re-written as:

$$\mathbf{Y}_c = \text{Softmax}\left(\alpha\frac{\pi_{\mathbf{c}}(\mathbf{Q})\pi_{\mathbf{c}}(\mathbf{K})^\top}{\sqrt{d}} \odot \mathbf{M}\right)\pi_{\mathbf{c}}(\mathbf{V})$$

$$= \text{Softmax}\left(\alpha\frac{\mathbf{P}_{\mathbf{c}}\mathbf{Q}(\mathbf{P}_{\mathbf{c}}\mathbf{K})^\top}{\sqrt{d}} \odot \mathbf{M}\right)\mathbf{P}_{\mathbf{c}}\mathbf{V}$$

$$= \text{Softmax}\left(\alpha\frac{\mathbf{P}_{\mathbf{c}}(\mathbf{Q}\mathbf{K}^\top)\mathbf{P}_{\mathbf{c}}^\top}{\sqrt{d}} \odot \mathbf{M}\right)\mathbf{P}_{\mathbf{c}}\mathbf{V} \tag{41}$$

by multiplying $\mathbf{P}_c\mathbf{P}_c^\top$ to both sides of $\mathbf{M}$ we have:

$$\mathbf{Y}_c = \text{Softmax}\left(\alpha\frac{\mathbf{P}_{\mathbf{c}}(\mathbf{Q}\mathbf{K}^\top)\mathbf{P}_{\mathbf{c}}^\top}{\sqrt{d}} \odot \mathbf{P}_c\mathbf{P}_c^\top\mathbf{M}\mathbf{P}_c\mathbf{P}_c^\top\right)\mathbf{P}_{\mathbf{c}}(\mathbf{V}) \tag{42}$$

$$= \text{Softmax}\left(\alpha\frac{\mathbf{P}_{\mathbf{c}}(\mathbf{Q}\mathbf{K}^\top)\mathbf{P}_{\mathbf{c}}^\top}{\sqrt{d}} \odot \mathbf{P}_c(\mathbf{P}_c^\top\mathbf{M}\mathbf{P}_c)\mathbf{P}_c^\top\right)\mathbf{P}_{\mathbf{c}}(\mathbf{V}) \tag{43}$$

Since the multiplication with the decay mask and the softmax operation are element-wise (i.e., applied row-wise for each query), the permutation matrices $\mathbf{P}_c$ and $\mathbf{P}_c^\top$ can be factored out of the attention computation. This results in the following expression:

$$\mathbf{Y}_c = \mathbf{P}_c\text{Softmax}\left(\alpha\frac{\mathbf{Q}\mathbf{K}^\top}{\sqrt{d}} \odot \mathbf{P}_c^\top\mathbf{M}\mathbf{P}_c\right)\underbrace{\mathbf{P}_c^\top\mathbf{P}_c}_{\mathbf{I}}\mathbf{V} = \mathbf{P}_c\text{Softmax}\left(\alpha\frac{\mathbf{Q}\mathbf{K}^\top}{\sqrt{d}} \odot \underbrace{\mathbf{P}_c^\top\mathbf{M}\mathbf{P}_c}_{\pi_c^{-1}(\mathbf{M})}\right)\mathbf{V} \tag{44}$$

Since the order of $\mathbf{Y}_c$ corresponds to the permuted input $\mathbf{X}_c$, we can recover the output in the original (basis) order by applying the inverse permutation, i.e., multiplying by $\mathbf{P}_c^\top$. Therefore, the final output $\widetilde{\mathbf{Y}}_{\mathbf{c}}$ aligned with the original input $\mathbf{X}$ is:

$$\widetilde{\mathbf{Y}}_{\mathbf{c}} = \mathbf{P}_c^\top\mathbf{Y}_c = \text{Softmax}\left(\alpha\frac{\mathbf{Q}\mathbf{K}^\top}{\sqrt{d}} \odot \mathbf{P}_c^\top\mathbf{M}\mathbf{P}_c\right)\mathbf{V} \tag{45}$$

This confirms that applying attention to a permuted input using the base decay mask is equivalent to applying attention to the original input with a permuted (reordered) decay mask $\pi_c^{-1}(\mathbf{M}) = \mathbf{P}_c^\top\mathbf{M}\mathbf{P}_c$.

$\square$

This proof is also visualized in Figure 6, illustrating that applying attention using a permuted decay mask based on curve $c$ (e.g., the snake curve in the figure) is equivalent to permuting the input sequence according to $c$, computing attention with the original decay mask defined in the basis curve (e.g., Z-curve in our study), and then reordering the output back to the original sequence order.

**Disclaimer** In practice, it is unnecessary to explicitly define a permutation function $\pi$ or construct a matrix $\mathbf{P}$. The reordering can be efficiently achieved by simply storing the corresponding indices. $\mathbf{P}$ and $\pi$ are used for mathematical clarity and formalism only.

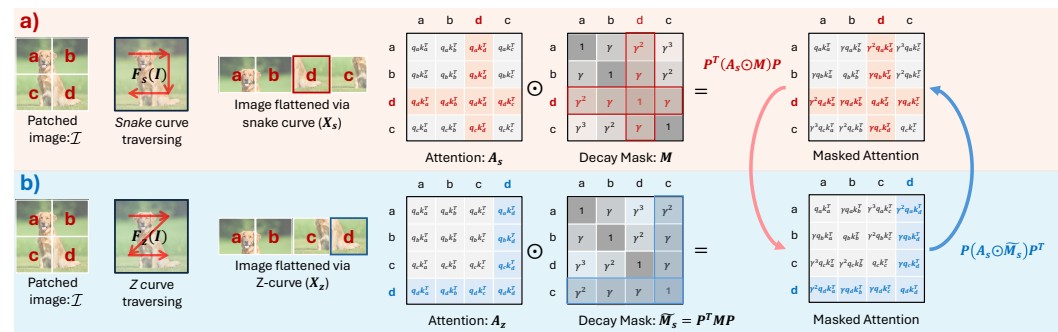

Figure 6: *Effect of SFCs on flattened Image:* Visually showing the equivalence between **a)** Permuting the input sequence according to $c$ (e.g., the snake curve) to get $\mathbf{X}_S$, multiplying the attention $\mathbf{A}_S$ with the original decay mask defined in the basis curve $\mathbf{M}$ (e.g., Z-curve in our study), and then reordering the output back to the original and **b)** Calculating attention $\mathbf{A}_z$ with basis curve ordered $\mathbf{X}_z$, using a permuted decay mask $\widehat{\mathbf{M}}_c$.

## D  FUTHER DESIGN DETAILS

In this section, we outline key design choices made in the implementation of VIOLIN models.

### D.1  INITIALIZATION

Since $\gamma_c = \text{sigmoid}(\beta_c)$ is exponentiated over the sequence length in the VIOLIN decay mask, it is important to initialize it close to 1, which is also highlighted in the Linear Transformer literature (Orvieto et al., 2023; Sun et al., 2023). For pretraining VIOLIN models, we initialize $\beta_c$ uniformly in the range $[5, 9]$, which corresponds to $\gamma_c \in (0.9820, 0.9998)$. This ensures that the initial mask values $\mathbf{M}_c[i, j] \in (0.03, 0.962)$ for $N = 196$, maintaining a stable and controlled decay. For numerical results on the effect of initialization, see Appendix E.4.

During full fine-tuning, we initialize the model using the pretrained baseline. In this setting, since the query/key/value weights $\mathbf{W_Q}, \mathbf{W_K}, \mathbf{W_V}$ are already trained during pretraining and VIOLIN attention is introduced and used only at fine-tuning, we initialize the scaling factor $\alpha$ using a Gaussian distribution centered at 1 to allow for smooth adaptation. For $\beta_c$, we use a uniform initialization in the range $[15, 20]$. This setup avoids a steep drop in attention scores while allowing the model to gradually adapt to the newly introduced decay mask $\mathbf{M}_{\text{VIOLIN}}$. All other initialization settings in VIOLIN exactly follow those of the original baselines without any modification.

*All other configurations, such as data augmentation, optimizer, initialization, model parameters, and training setups are kept exactly the same as in the original baselines, with no modifications.*

### D.2  ADAPTATION OF VIOLIN TO VARIOUS ARCHITECTURES

VIOLIN attention supports both the use of a classification token and Global Average Pooling (GPA) (Lin et al., 2013; Lu et al., 2022). For pretraining of DeiT models, we remove the classification token and instead apply Global Average Pooling (GAP). The attention module is replaced with VIOLIN attention, while the rest of the model, including positional embeddings, layer normalization, and other components, remains unchanged, see Appendix E.5 for details. For fine-tuning the classification token remains intact.

In the DINO setting, both teacher and student models are initialized with VIOLIN attention, with all other weights handled as usual. Due to the multi-crop training, the attention module encounters varying sequence lengths. However, since the construction of $\mathbf{M}_{\text{VIOLIN}}$ naturally adapts to any sequence length, this poses no issue.

To accommodate the classification token, we modify the corresponding rows and columns of $\mathbf{M}_{\text{VIOLIN}}$ by setting $\gamma_{\text{cls}} = 1$. We also experimented with a learnable $\gamma_{\text{cls}} \in [0, 1]$ but observed no significant performance gains. The rest of the model structure follows the original DINO architecture.

**VIOLIN with hierarchical and convolutional architectures**  Hierarchical transformer architectures such as Swin (Liu et al., 2021) and convolutional-transformer hybrids like PVT (Wang et al., 2021) differ fundamentally from vanilla ViTs in how attention is computed. Instead of applying full attention across the entire sequence, they restrict the receptive field by using windowed or spatially localized attention, often combined with hierarchical feature maps. This design introduces locality explicitly into the architecture, reducing the need for additional spatial priors such as those provided by SFCs.

In such settings, applying SFC-guided decay masks becomes problematic for two main reasons. First, SFCs are meaningful when attention spans the *entire* sequence of image patches, since the curve defines a global traversal order. In hierarchical models, however, attention is restricted to local windows or pyramid levels, where the notion of a global SFC ordering no longer applies. Second, many of these architectures already incorporate inductive biases (through localized windows, shifting strategies, or convolutional layers), so introducing additional SFC-based priors could interfere with rather than complement their design.

Thus, VIOLIN is best suited for standard ViTs and related architectures where attention is fully global, the sequence is flattened in a fixed order (commonly the Z-curve), and inductive biases are otherwise minimal. In contrast, hierarchical or convolutional variants already bake spatial priors directly into their architecture, making SFC-based masking redundant or ill defined.

Consistent with our analysis, when we integrated VIOLIN into Swin at tiny and small scales during pretraining, we achieved minimal accuracy improvements of $0.2\%$ and $0.1\%$, respectively, as shown in Table 10. The VIOLIN mask is applied at every stage and layer, with each mask being independently learned and unique to its respective layer. The remaining architecture follows the original Swin model structure.

Table 10: *Pretraining of Swin models:* The performance of baseline model is compared against VIOLIN for ImageNet pretraining. Changes with respect to the baseline are shown inside $(\cdot)$ next to the accuracies.

| Model | Top-1 Accuracy (%) | |
| | Baseline | VIOLIN |
| --- | --- | --- |
| Swin-T | 81.3 | **81.5** (+0.2) |
| Swin-S | 83.0 | **83.1** (+0.1) |

**VIOLIN with video transformers**  Video transformers operate on spatiotemporal tokens, and VIOLIN can be incorporated into these models in a straightforward way because it only rescales the attention scores between tokens. This makes VIOLIN orthogonal to additional mechanisms used in video models, such as the dual masking strategy in VideoMAE V2 (Wang et al., 2023).

There are two natural ways to extend VIOLIN :

1. **Spatial-only SFCs (2D per frame).** The same 2D SFCs used for images can be applied independently to the $(H, H)$ grid of each frame, while keeping the temporal dimension unchanged. This provides a per-frame spatial prior and mirrors the image setting.

2. **Full spatiotemporal SFCs (3D).** Following definition 2.1, SFCs naturally generalize to arbitrary dimensions. Thus, we can define 3D SFCs over the full $(T, H, W)$ grid (e.g., 3D Hilbert or 3D Morton curves) and compute distances based on each token's original spatiotemporal position. The resulting decay masks encourage locality across both space and time. Masks can be computed once over the full grid and then indexed to the visible token subset, analogous to how positional embeddings are handled in VideoMAE.

Both approaches are fully compatible with video MAE-style training: they require no changes to masking or reconstruction objectives, they can be applied to both encoder and decoder, and they provide a meaningful structural prior, especially under high masking ratios where positional structure becomes crucial.

Overall, extending VIOLIN to video models is a promising direction for future work, as spatiotemporal SFCs may offer strong inductive bias with minimal additional cost.

# E ABLATION STUDIES

In this section, we provide comprehensive ablation studies on various elements of VIOLIN . For all ablations, we utilize different scales of DeiT models and we keep the training recipe the same. We use a patch size of 16 and a resolution $224 \times 224$ for each one of the models.

## E.1 POSITIONAL EMBEDDINGS

To evaluate the impact of positional embeddings, we pretrain the VIOLIN DeiT-B model both with and without them, see Table 11. The results indicate that positional embeddings provide a performance boost, leading us to retain the original positional embedding configurations of the base models.

Table 11: *Ablation on positional embeddings (PE):* The performance of the baseline model with PE is compared against VIOLIN with (w) and without (wo) PE. Changes with respect to the baseline are shown inside $(\cdot)$ next to the accuracies.

| Model | Top-1 Accuracy (%) | | |
|---|---|---|---|
| | Baseline | VIOLIN w PE | VIOLIN wo PE |
| DeiT-B | 81.8 | **81.9 (+0.1)** | 81.5 (-0.3) |

## E.2 ALTERNATIVE CURVE CONFIGURATIONS

We examine the individual contribution of each curve to the overall performance. To do so, we pretrain DeiT-S using all possible combinations of the four curves, resulting in $2^4 = 16$ variations. The accuracies of each configuration are presented in Table 12. Note that whenever a curve has is used, the transposed version is also included. In other words, if the snake curve is included, its transposed variant $\text{Snake}^\top$ is also utilized.

The results reveal that while certain curve combinations yield more substantial improvements than others, each curve contributes meaningfully to the overall performance. Thus, we retain all four curves in the VIOLIN configuration, leveraging their complementary spatial information.

We further analyze the learned decay parameters $\gamma_c$ for DeiT-B in Figure 7, observing that most remain close to one, indicating active use of long-range spatial information. Smaller values act as implicit curve selection, as these decay masks would contribute to the average minimally, with certain layers and heads emphasizing particular curves.

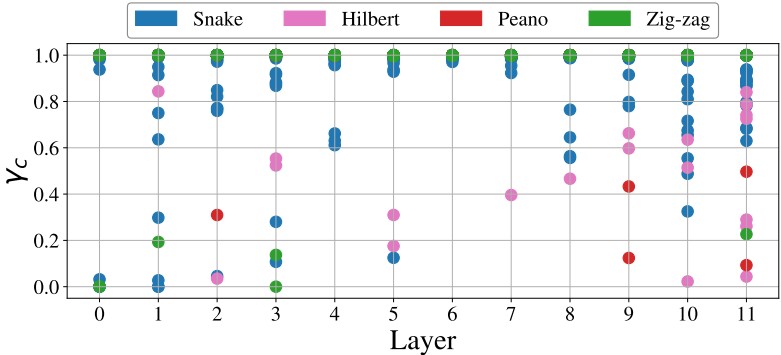

Figure 7: $\gamma_c$ *values:* $\gamma_c$ values of VIOLIN DeiT-B model are presented across layers, heads and curves. Most remain close to one, indicating active use of long-range spatial information.

Additionally, we explore several alternative configurations, as detailed in Table 13. For instance, we evaluate the use of only the four original curves referred as $\mathcal{C}_{\text{normal}}$ (snake, zig-zag, Hilbert, and Peano) and only their rotated counterparts $\mathcal{C}_{\text{transposed}}$ (snake$^\top$, zig-zag$^\top$, Hilbert$^\top$, and Peano$^\top$). We also test using only the default Z-curve ordering, which results in a $0.7\%$ accuracy gain.

Moreover, we define relative distances using a Manhattan mask, inspired by RMT (Fan et al., 2024). Lastly, we experiment with a set of randomized SFCs, where the flattened image is shuffled with a random fixed order across all layers and heads. This model fails to converge to a meaningful accuracy.

Table 12: *Ablation on the effect of each curve:* The performance of the baseline model is compared against VIOLIN with different curve combinations. ✔ indicates the curse is in the set, whereas ✗ means it is not. Changes with respect to the baseline are shown inside $(\cdot)$ next to the accuracies.

| Model | Snake Curve | Zig-Zag Curve | Hilbert Curve | Peano Curve | Top-1 Acc (%) |
|---|---|---|---|---|---|
| DeiT-S (Baseline) | ✗ | ✗ | ✗ | ✗ | 79.9 |
| | ✔ | ✗ | ✗ | ✗ | 80.0 (+0.1) |
| | ✗ | ✔ | ✗ | ✗ | 80.2 (+0.3) |
| | ✗ | ✗ | ✔ | ✗ | 79.9 — |
| | ✗ | ✗ | ✗ | ✔ | 80.4 (+0.5) |
| | ✔ | ✔ | ✗ | ✗ | 80.3 (+0.4) |
| | ✔ | ✗ | ✔ | ✗ | 80.4 (+0.5) |
| | ✔ | ✗ | ✗ | ✔ | 80.3 (+0.4) |
| | ✗ | ✔ | ✗ | ✗ | 80.3 (+0.4) |
| | ✗ | ✔ | ✗ | ✔ | 80.5 (+0.6) |
| | ✗ | ✗ | ✔ | ✔ | 80.2 (+0.3) |
| | ✔ | ✔ | ✔ | ✗ | 80.4 (+0.5) |
| | ✔ | ✔ | ✗ | ✔ | 80.4 (+0.5) |
| | ✔ | ✗ | ✔ | ✔ | 80.5 (+0.6) |
| | ✗ | ✔ | ✔ | ✔ | 80.5 (+0.6) |
| VIOLIN DeiT-S (Ours) | ✔ | ✔ | ✔ | ✔ | **80.7 (+0.8)** |

This further emphasizes the importance of a *structured* SFC as the unstructured curves do not allow model to capture meaningful information from the data.

### E.3 ALTERNATIVE MASKING STRATEGIES

Another critical design choice is the masking strategy. We compare VIOLIN , which follows the structure $S(\mathbf{A}' \odot \mathbf{M})$, where $S$ denotes the row-wise softmax operation, $\mathbf{A}' = \alpha \frac{\mathbf{Q}\mathbf{K}^\top}{\sqrt{d}}$, and $\mathbf{M} = \mathbf{M}_{\text{VIOLIN}}$ for a cleaner notation. Our findings indicate that the $S(\mathbf{A}' \odot \mathbf{M})$ configuration outperforms all other masking alternatives.

### E.4 OTHER DESIGN ELEMENTS

Furthermore, in Table 15, we illustrate the impact of additional design choices described in Appendix D, such as initialization and the scaling parameter $\alpha$. Additionally, we assess the effect of fixing $\gamma_c$ at a constant value of 0.9996 instead of learning it. The results indicate that proper initialization and a learnable $\gamma_c$ are essential for achieving accuracy gains, while the scaling parameter $\alpha$ primarily contributes to training stability, particularly in larger models.

### E.5 GLOBAL AVERAGE POOLING (GAP)

Considering the output of the attention mechanism for each token in the last layer, we can write

$$\mathbf{y}_i = \sum_{j=1}^{N} \frac{\exp(\mathbf{q}_i^\top \mathbf{k}_j)}{\sum_{j'=1}^{N} \exp(\mathbf{q}_i^\top \mathbf{k}_{j'})} \mathbf{v}_j. \tag{46}$$

When the classification (CLS) token is used, the sequrnce length becomes $N+1$ where the first token is the CLS. When comparing the use of a global average pooling (GAP) (Lin et al., 2013; Lu et al.,

Table 13: *Ablation on different curve configurations:* The performance of the baseline model is compared against VIOLIN with different curve configurations: only original curves ($\mathcal{C}_{\text{normal}}$), only transposed curves ($\mathcal{C}_{\text{transposed}}$), only Z-curve, Manhattan distance-based mask and random curves. Changes with respect to the baseline are shown inside $(\cdot)$ next to the accuracies.

| Model | Top-1 Accuracy (%) | | | | | | |
|---|---|---|---|---|---|---|---|
| | Baseline | VIOLIN | $\mathcal{C}_{\text{normal}}$ | $\mathcal{C}_{\text{transposed}}$ | Z-curve | Manhattan | Random |
| DeiT-S | 79.8 | **80.7 (+0.9)** | 80.3 (+0.5) | 80.4 (+0.6) | 80.5 (+0.7) | 80.4 (+0.6) | ✗ |

Table 14: *Ablation on masking strategies:* The performance of the baseline model is compared against VIOLIN with different masking methods: $S(\mathbf{M}+\mathbf{A}')$, $S(\mathbf{A}')+\mathbf{M}$, $S(\mathbf{A}')\odot\mathbf{M}$, and $S(\mathbf{A}'\odot(\mathbf{I}+\mathbf{M}))$. Changes with respect to the baseline are shown inside $(\cdot)$ next to the accuracies.

| Model | Top-1 Accuracy (%) | | | | | |
|---|---|---|---|---|---|---|
| | Baseline | VIOLIN | $S(\mathbf{M}+\mathbf{A}')$ | $S(\mathbf{A}')+\mathbf{M}$ | $S(\mathbf{A}')\odot\mathbf{M}$ | $S(\mathbf{A}'\odot(\mathbf{I}+\mathbf{M}))$ |
| DeiT-S | 79.8 | **80.7 (+0.9)** | 80.1 (+0.3) | 80.5 (+0.7) | 80.5 (+0.7) | 79.1 (-0.7) |

Table 15: *Ablation on other elements of* VIOLIN *:* The performance of the baseline model is compared against VIOLIN with and without certain design elements: initialization, scaling factor $\alpha$ and learned $\gamma_c$. ✔ indicates it is included in the model, whereas ✗ means it is not. Changes with respect to the baseline are shown inside $(\cdot)$ next to the accuracies.

| Model | Initialization | Scaling | Learned $\gamma_c$ | Top-1 Acc (%) |
|---|---|---|---|---|
| DeiT-S (Baseline) | ✗ | ✗ | ✗ | 79.9 |
| | ✗ | ✔ | ✔ | 80.0 (+0.1) |
| | ✔ | ✗ | ✔ | **80.7 (+0.8)** |
| | ✔ | ✔ | ✗ | 80.3 (+0.4) |
| VIOLIN DeiT-S (Ours) | ✔ | ✔ | ✔ | **80.7 (+0.8)** |

2022) head versus a CLS head with a decay mask, the attention outputs are extracted as follows

$$\mathbf{y}_{\text{CLS}} = \sum_{j=1}^{N+1} \frac{\exp\big((\mathbf{q}_{CLS}^\top \mathbf{k}_j)\mathbf{M}[CLS,j]\big)}{\sum_{j'=1}^{N+1} \exp\big((\mathbf{q}_{CLS}^\top \mathbf{k}_{j'})\mathbf{M}[CLS,j']\big)} \, \mathbf{v}_j, \tag{47}$$

$$\mathbf{y}_{\text{GAP}} = \frac{1}{N} \sum_{i=1}^{N} \sum_{j=1}^{N} \frac{\exp\big((\mathbf{q}_i^\top \mathbf{k}_j)\mathbf{M}[i,j]\big)}{\sum_{j'=1}^{N} \exp\big((\mathbf{q}_i^\top \mathbf{k}_{j'})\mathbf{M}[i,j']\big)} \, \mathbf{v}_j. \tag{48}$$

As shown, in the case of the CLS token, the model only requires the attention distribution and relative distances with respect to the CLS token. In our setup, this reduces to $\mathbf{M}[CLS,j] = 1$, (or a a learned parameter $\beta_{CLS}$. By contrast, the GAP formulation is more expressive, as it aggregates attention information across all tokens. Importantly, the inclusion of the relative distance decay mask $\mathbf{M}[i,j]$ for all tokens makes GAP more effective in constructing the final representation. Therefore, similar to Vision SSMs such as Vision LSTM and Hydra (Alkin et al., 2024; Hwang et al., 2024), pooling-based outputs align naturally with spatially informed attention. Note that this calculations holds for last layer only, the remaining layers utilizes the mask fully.

VIOLIN attention supports both the use of a classification token and GPA. To assess the role of the classification token versus GAP with the VIOLIN mask, we pretrain all three scales of DeiT and report results in Table 16. While GAP often yields slightly better compatibility with VIOLIN , the improvements cannot be attributed to pooling alone, the gains are additive.

Most importantly, VIOLIN is *not dependent on GAP*. In DINO pretraining and VTAB-1K fine-tuning, where the cls_token is retained, VIOLIN still improves performance. This confirms that the benefits arise from the spatial priors introduced by VIOLIN , not from the choice of pooling strategy.

Table 16: *Ablation on GAP:* The performance of baseline model and VIOLIN is compared when they both have CLS or uses GAP. Baseline[†] indicates results taken from Chu et al. (2023). Changes with respect to the baseline, original model with CLS, are shown inside $(\cdot)$ next to the accuracies.

| Model | Top-1 Accuracy (%) | | | |
|---|---|---|---|---|
| | CLS | | GAP | |
| | Baseline | VIOLIN | Baseline[†] | VIOLIN |
| DeiT-T | 72.2 | 72.3 (+0.2) | 72.6 | **73.0 (+0.8)** |
| DeiT-S | 79.8 | 80.1 (+0.3) | 80.2 | **80.7 (+0.9)** |
| DeiT-B | 81.8 | 79.0 (-1.8) | - | **81.9 (+0.1)** |

# F ADDITIONAL RESULTS

## F.1 PRETRAINING OF LARGER MODELS

As discussed in Appendix B.1, when both model capacity and training data are sufficiently large, ViTs can implicitly learn spatial biases directly from data. In such scenarios, the relative contribution of VIOLIN is naturally smaller, as seen in the DeiT and DINO base scale pretraining results in Table 17, which show only marginal gains. This is expected and lies beyond the primary scope of our work, which focuses on small models and data-scarce settings where inductive biases are most impactful.

It is important to note that smaller gains at scale do not diminish the relevance of VIOLIN for larger models. In fact, our fine-tuning experiments (Section 4.1, Table 18) demonstrate that when data is limited, spatial priors provided by VIOLIN substantially improve performance, even for models with hundreds of millions of parameters. This highlights that VIOLIN remains valuable in practice, not by competing with scale, but by enhancing efficiency and adaptability in data-constrained regimes.

Table 17: *Pretraining results of larger models on ImageNet-1K*: Comparison of the top-1 accuracies of baseline models with their VIOLIN counterparts. The values in parentheses (·) indicate the accuracy difference compared to the baseline. The best performance between each pair of models is highlighted in **bold**. For DINO models, both KNN and linear probe evaluations are reported and (300) indicate the number of training epochs. **(Left)** Supervised, **(Right)** Self-supervised training.

| Model | # Param. | Top-1 Accuracy (%) Baseline | VIOLIN |
|---|---|---|---|
| DeiT-B | 86M | 81.8 | **81.9 (+0.1)** |

| Model | | # Param. | Top-1 Accuracy (%) Baseline | VIOLIN |
|---|---|---|---|---|
| DINO-B (300) | KNN | 86M | 76.1 | **76.1 (——)** |
| | Linear | | 78.2 | **78.4 (+0.2)** |

## F.2 FINE-TUNING OF VIOLIN PRETRAINED MODELS

We fine-tune the VIOLIN DeiT, and DINO pretrained models from Section 4.2 and Appendix F.1 on the VTAB-1K dataset. The accuracies for each category and the overall average are presented in Table 18, alongside the baseline accuracies of the baseline fine-tuned models. We observe that VIOLIN increases the performance across all models and scales compared to original baselines. DeiT,and DINO models achieve impressive improvements of up to 1.92% with up to 2.87% improvement in individual categories. We note that similar to Table 3 in this setting, Structured group shows the highest accuracy gain. This further shows the broad applicability of VIOLIN , enhancing diverse architectures with close to zero computational overhead.

Notably, we compare Table 3 and Table 18, fine-tuning with an mask learned only during fine-tuning for all models yields better performance in different tasks compared to pretraining with it. We hypothesize that this is because the model starts with generic pretrained representations and gains additional flexibility by learning spatial structure tailored specifically to the downstream task. This is particularly advantageous when the target task differs substantially from the pretraining domain.

Table 18: *Fine-tuning results on VTAB-1K (Setting 2)*: Comparison of the top-1 accuracies of baseline models and their pretrained VIOLIN counterparts across the VTAB-1K benchmark. The three task groups are abreviated as NAT. = Natural, SPE. = Specialized, and STR. = Structured. The values in parentheses (·) indicate the accuracy difference compared to the baseline. The best performance within each model pair is highlighted in **bold**.

| Model | Param. | Top-1 Accuracy (%) | | | | | | | |
|---|---|---|---|---|---|---|---|---|---|
| | | Baseline | | | | VIOLIN | | | |
| | | NAT. | SPE. | STR. | Avg. | NAT. | SPE. | STR. | Avg. |
| DeiT-T | 5M | 69.56 | 82.34 | 53.57 | 65.52 | **70.71 (+1.15)** | **82.64 (+0.30)** | **54.52 (+0.95)** | **66.41 (+0.89)** |
| DeiT-S | 22M | 73.64 | 84.30 | 53.44 | 67.38 | **75.24 (+1.60)** | **84.87 (+0.57)** | **56.31 (+2.87)** | **69.30 (+1.92)** |
| DeiT-B | 86M | **76.93** | **85.52** | 57.00 | 70.35 | 76.54 (-0.39) | 85.44 (-0.08) | **58.90 (+1.90)** | **70.99 (+0.64)** |
| DINO-S | 22M | 75.35 | 85.09 | **60.65** | 71.21 | **76.29 (+0.94)** | **85.75 (+0.66)** | 60.61 (-0.04) | **71.68 (+0.47)** |
| DINO-B | 86M | 77.50 | 85.77 | 58.47 | 71.23 | **77.82 (+0.32)** | **85.83 (+0.06)** | **58.77 (+0.30)** | **71.49 (+0.26)** |

### F.3 MULTI-RESOLUTION CLASSIFICATION

Following Heo et al. (2024), we test the resolution scalability of VIOLIN models. We present the top-1 accuracies for DeiT-S, and DeiT-B models across input resolutions ranging from 144 to 512 in Figure 8. We use bicubic interpolation for all positional embeddings (Heo et al., 2024). In the top plot, we observe that although VIOLIN without positional embeddings performs slightly worse than the baseline at the training resolution (224), it begins to outperform the baseline at higher resolutions. In the second and third plots, where VIOLIN is combined with positional embeddings, for most resolutions, VIOLIN preserves or expands the performance gap compared to baselines. These results suggest that the decay mask used in VIOLIN generalizes effectively to higher resolutions, making it a resolution-robust enhancement for ViTs.

Another interesting application of context extrapolation is video understanding. Following Caron et al. (2021), we generate a segmentation video using VIOLIN DINO-B model. While the training resolution is 224, for video, VIOLIN extends to $768 \times 432$ resolution. Some frames are provided in Figure 9 and the full video can be found in our GitHub repository.

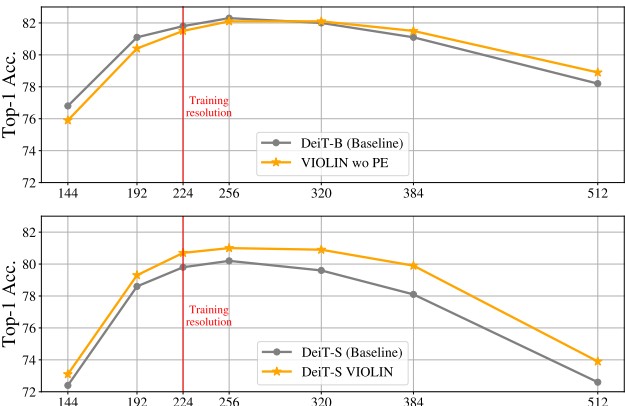

Figure 8: *Resolution expansion:* Top-1 accuracies of DeiT-B (top), DeiT-S (middle) and DeiT-III-S (bottom) models and their VIOLIN counterparts at different resolutions on ImageNet. Training resolution of 224 is highlighted in red.

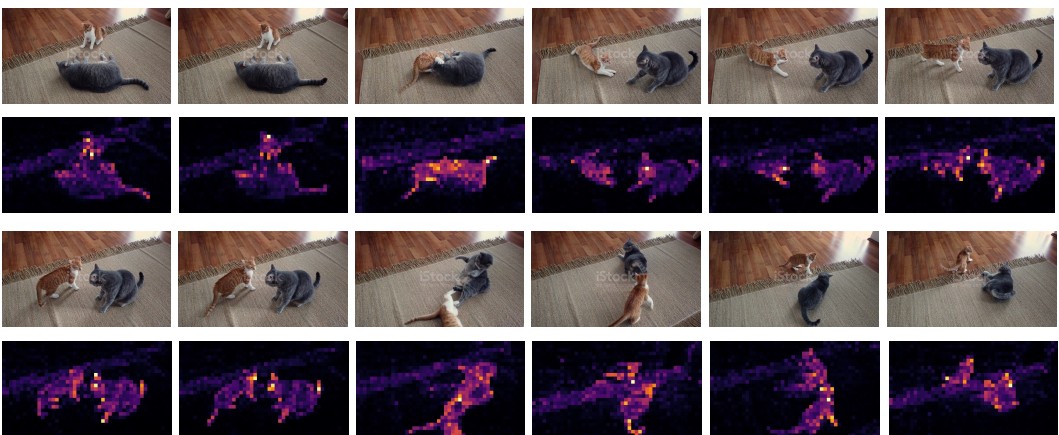

Figure 9: *Video understanding*: Frame by frame video understanding of VIOLIN -DINO in base scale. The full video and generation codes are also included in the github repository of VIOLIN .

## F.4 ADDITIONAL VISUALIZATIONS

In Figure 10, we present the 1D flattened sequences of the patched image **(a)**, corresponding to the curves illustrated in Figure 1. Figure 12 compares attention heatmaps of DeiT and VIOLIN models, fine-tuned on Structured group datasets. Figure 13 visualizes the attention heatmaps of the VIOLIN DeiT-B model using various images. We adopt the average diagonal visualization strategy as proposed in (Liu et al., 2024b). Additionally, in Figure 11 we visualize the mask pattern for a middle pixel under the snake curve for different values of $\gamma$. As expected, when $\gamma \approx 1$, the head attends broadly across the entire image, whereas smaller $\gamma$ values produce a much more localized receptive field, emphasizing spatial neighbors.

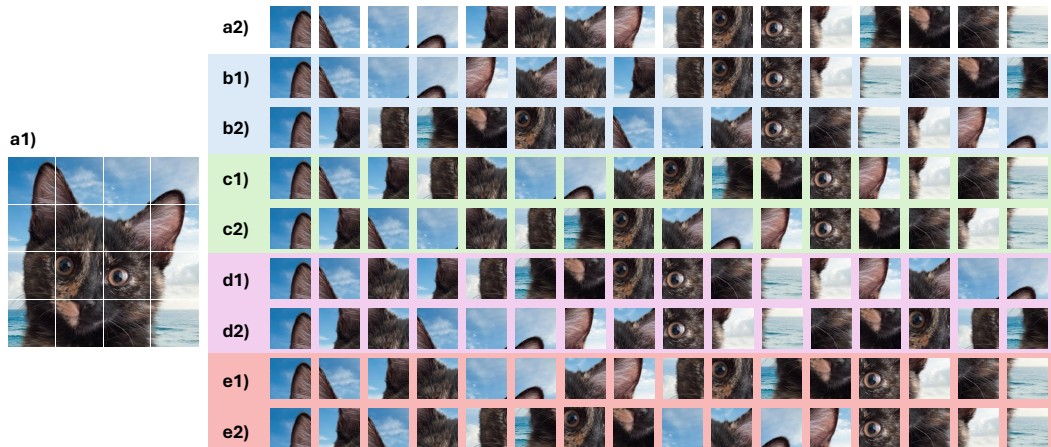

Figure 10: *Flattened Space Filling Curve paths:* Examples of flattened images with different traversal paths followed in VIOLIN . **(a1)** Original patchedimage. **(a2)** Z-curve **(b1)** Snake curve, **(b2)** Transposed Snake curve, **(c1)** Zig-zag curve, **(c2)** Transposed Zig-zag curve, **(d1)** Hilbert curve, **(d2)** Transposed Hilbert curve, **(e1)** Peano curve, **(e2)** Transposed Peano curve.

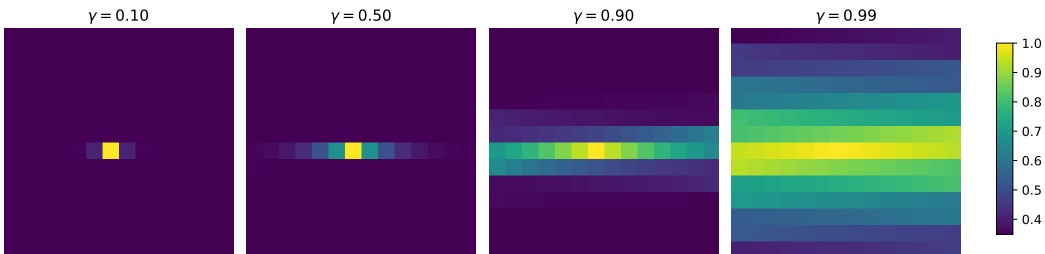

Figure 11: *Effect of $\gamma$ on the decay mask:* Visualization of the decay mask for a central pixel under the Snake curve for different values of $\gamma$. Larger $\gamma$ values yield more global attention, while smaller $\gamma$ restrict the effective receptive field to local regions.

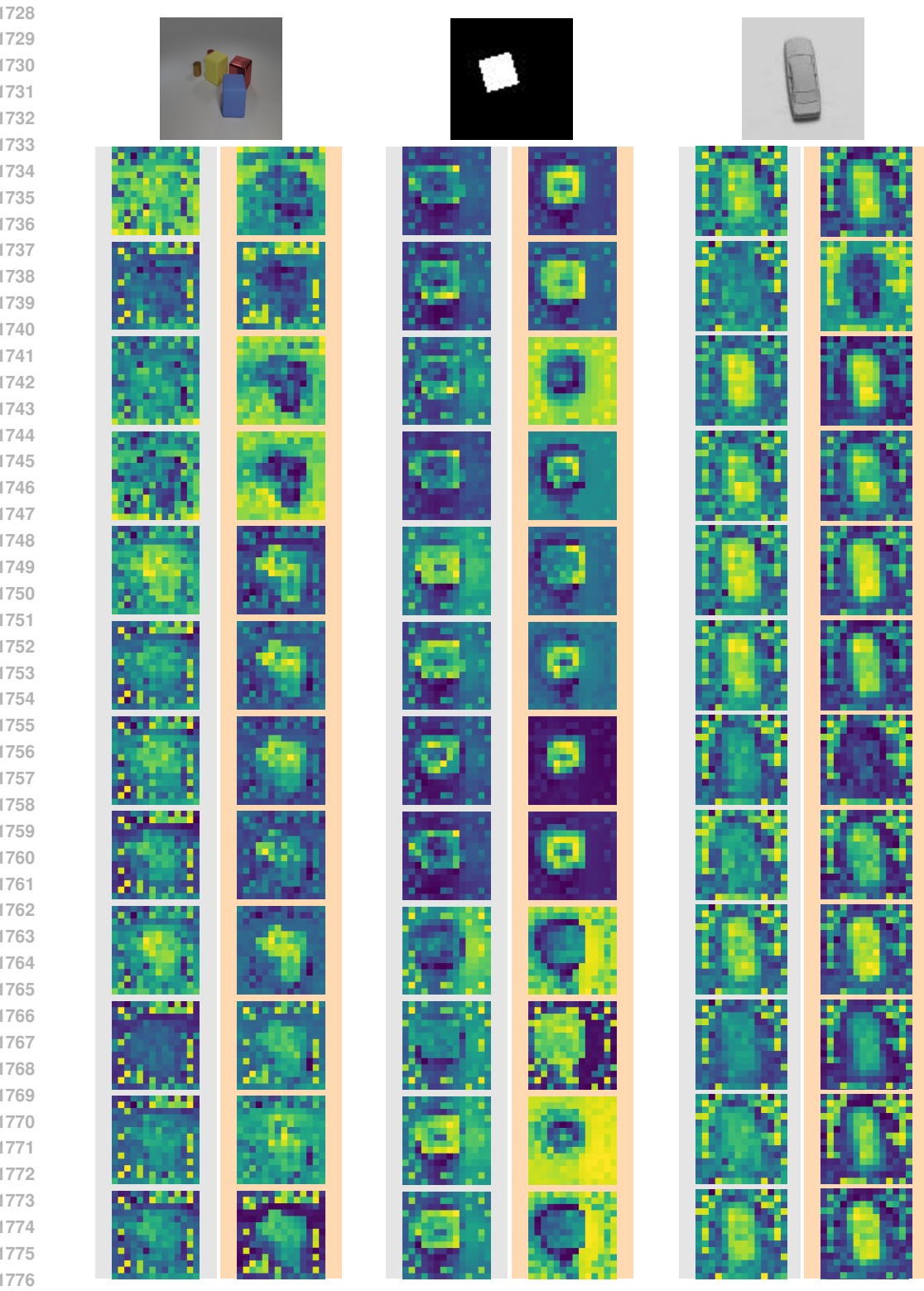

Figure 12: *Attention heatmaps on Structured tasks:* Examples are taken from three datasets in the Structured group: CLEVR-Count, dSprites-Location, and SmallNORB-Azimuth. We compare attention scores of DeiT-B (left) and VIOLIN (right), fine-tuned on the corresponding dataset. Visualizations are from layer 12, with rows showing heads 112. Since both models share the same pretrained initialization, attention heads are initially identical before fine-tuning. After fine-tuning, VIOLIN produces more accurate and focused heads, with better object coverage and more uniform color outside the objects, indicating reduced attention to irrelevant regions.

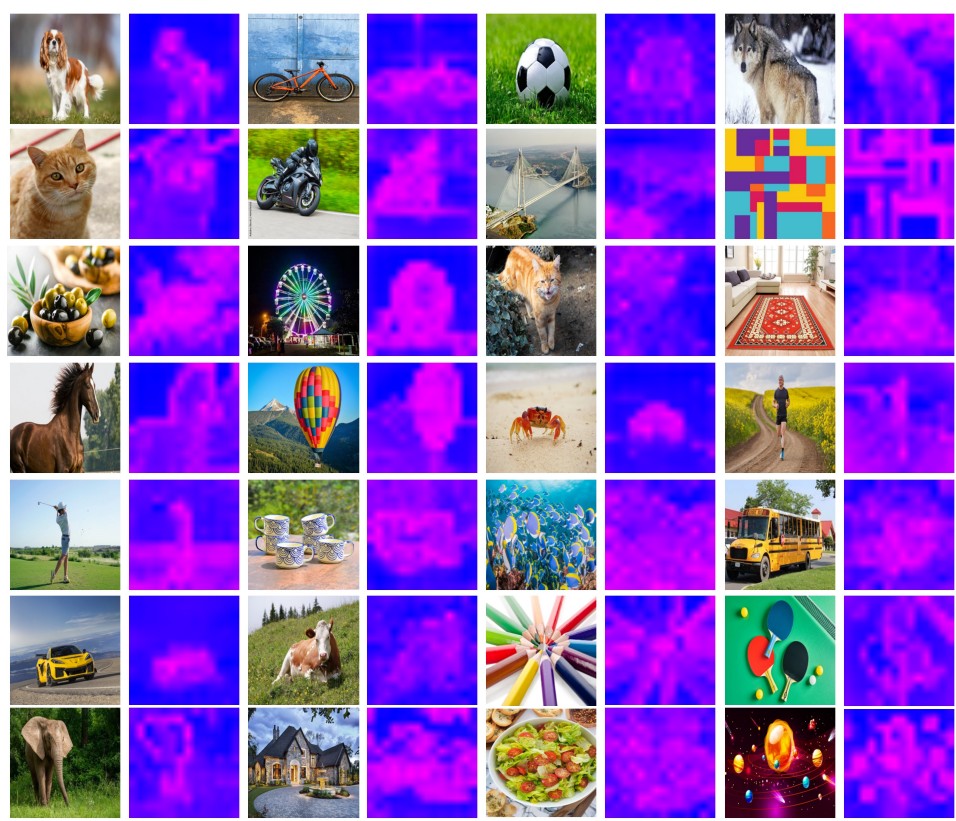

Figure 13: *Attention heatmap visualization of* VIOLIN *DeiT-B*: The average diagonal of the masked attention is visualized followed by (Liu et al., 2024b).

## F.5 DETAILS AND INDIVIDUAL RESULTS ON VTAB-1K DATASET

VTAB (Zhai et al., 2019) contains 19 tasks which cover a broad spectrum of domains and semantics that are grouped into three sets: NATURAL, SPECIALIZED, and STRUCTURED.

The NATURAL group represents natural images and classical vision problems. The group includes Caltech101, CIFAR-100, DTD, Flowers102, Pets, Sun397, and SVHN datasets.

The SPECIALIZED group also contains images of the world, but they are captured through specialist equipment. These images have different invariances to those in the NATURAL tasks. It includes Resisc45 and EuroSAT, Patch Camelyon, and Diabetic Retinopathy datasets.

The STRUCTURED group assesses comprehension of the structure of a scene, for example, object counting, or 3D depth prediction. Most of the tasks are generated from simulated environments, whose structure is easy for a human, but their domain differs greatly to datasets like ImageNet. It includes Clevr count and distance, dSprites location and orientation, SmallNORB, DMLab, and KITTI. In Tables 19 to 21, we present the accuracy scores of each model on all VTAB-1K datasets.

Table 19: *VTAB Results-Natural Subset:* Individual scores for each dataset.

| | Model | CIFAR | Caltech101 | DTD | Flowers102 | Pets | SVHN | Sun397 |
|---|---|---|---|---|---|---|---|---|
| Natural | DeiT-T | 48.36 | 86.9 | 63.97 | 86.43 | 87.14 | 78.28 | 35.87 |
| | VIOLIN DeiT-T | 51.21 | 86.48 | 64.75 | 87.24 | 86.77 | 83.16 | 35.38 |
| | DeiT-T $\odot M_{VIOLIN}$ | 51.17 | 87.8 | 65.43 | 89.17 | 86.75 | 85.78 | 37.17 |
| | DeiT-S | 57.38 | 89.06 | 68.83 | 91.09 | 91.13 | 75.82 | 42.19 |
| | VIOLIN DeiT-S | 60.71 | 88.06 | 68.33 | 91.12 | 91.19 | 85.38 | 41.93 |
| | DeiT-S $\odot M_{VIOLIN}$ | 59.6 | 89.78 | 69.08 | 92.5 | 91.89 | 86.15 | 43.45 |
| | DeiT-B | 61.38 | 90.33 | 69.06 | 93.73 | 92.43 | 85.95 | 45.59 |
| | VIOLIN DeiT-B | 63.32 | 89.55 | 68.37 | 92.1 | 92.04 | 86.22 | 44.15 |
| | DeiT-B $\odot M_{VIOLIN}$ | 61.99 | 91.07 | 70.14 | 93.97 | 92.75 | 90.22 | 45.56 |
| | DeiT-B LoRA | 62.37 | 90.07 | 69.27 | 93.26 | 92.3 | 90.58 | 44.35 |
| | DeiT-B $\odot M_{VIOLIN}$ LoRA | 65.36 | 90.92 | 70.62 | 93.57 | 92.37 | 91.86 | 45.19 |
| | DeiT-B DoRA | 63.81 | 90.78 | 69.29 | 91.79 | 89.95 | 88.75 | 44.12 |
| | DeiT-B $\odot M_{VIOLIN}$ DoRA | 66.38 | 90.97 | 69.82 | 92.77 | 91.71 | 90.26 | 44.64 |
| | DeiT-III-S | 59.08 | 88.53 | 67.09 | 91.13 | 91.85 | 84.65 | 43.57 |
| | DeiT-III-S $\odot M_{VIOLIN}$ | 62.18 | 88.78 | 69.4 | 93.92 | 91.35 | 89.98 | 43.6 |
| | DeiT-III-B | 64.39 | 89.56 | 70.8 | 94.63 | 93.38 | 87.28 | 47.28 |
| | DeiT-III-B $\odot M_{VIOLIN}$ | 66.77 | 89.97 | 71.38 | 95.53 | 93.61 | 91.24 | 46.19 |
| | DeiT-III-L | 65.16 | 87.89 | 71.58 | 94.39 | 93.23 | 71.17 | 48.65 |
| | DeiT-III-L $\odot M_{VIOLIN}$ | 66.74 | 87.67 | 72.34 | 95.01 | 93.28 | 78.7 | 48.58 |
| | DeiT-III-H | 64.34 | 88.2 | 71.22 | 94.95 | 92.96 | 68.76 | 48.46 |
| | DeiT-III-H $\odot M_{VIOLIN}$ | 65.16 | 88.18 | 71.35 | 95.18 | 93.33 | 72.72 | 48.7 |
| | DINO-S | 54.32 | 93.95 | 68.12 | 91.28 | 88.62 | 90.24 | 40.93 |
| | VIOLIN DINO-S | 56.05 | 91.95 | 69.33 | 95.26 | 89.62 | 91.65 | 40.2 |
| | DINO-S $\odot M_{VIOLIN}$ | 57.38 | 90.92 | 68.88 | 95.18 | 89.44 | 90.61 | 41.45 |
| | DINO-B | 58.57 | 93.7 | 70.64 | 95.84 | 90.21 | 89.69 | 43.86 |
| | VIOLIN DINO-B | 59.96 | 92.13 | 71.84 | 95.69 | 90.49 | 90.78 | 43.83 |
| | DINO-B $\odot M_{VIOLIN}$ | 62.21 | 93.32 | 71.58 | 96.1 | 90.74 | 91.74 | 44.87 |

Table 20: *VTAB Results-Structured Subset:* Individual scores for each dataset. SN refers to Small-Norm, and dS represents dSprites.

| | Model | CLEVR Count | CLEVR Dist | DMLab | KITTI | dS Loc | dS Ori | SN Azi | SN Ere |
|---|---|---|---|---|---|---|---|---|---|
| **Structured** | DeiT-T | 71.37 | 60.37 | 44.26 | 78.81 | 69.04 | 41.86 | 30.28 | 32.57 |
| | VIOLIN DeiT-T | 72.73 | 61.7 | 47.98 | 79.7 | 68.7 | 46.11 | 25.31 | 33.96 |
| | DeiT-T ⊙M$_{VIOLIN}$ | 74.41 | 59.84 | 46.37 | 80.78 | 78.32 | 50.91 | 31.33 | 38.05 |
| | DeiT-S | 75.08 | 58.15 | 45.74 | 78.43 | 63.3 | 48.13 | 26.24 | 32.48 |
| | VIOLIN DeiT-S | 78.26 | 59.25 | 49.91 | 81.29 | 64.63 | 53.16 | 27.37 | 36.59 |
| | DeiT-S ⊙M$_{VIOLIN}$ | 78.87 | 59.2 | 50.59 | 80.4 | 73.52 | 53.44 | 32.48 | 37.62 |
| | DeiT-B | 79.01 | 60.1 | 47.03 | 82.61 | 66.7 | 53.38 | 30.87 | 36.32 |
| | VIOLIN DeiT-B | 82.6 | 61.72 | 52.84 | 80.97 | 68.44 | 55.47 | 31.72 | 37.45 |
| | DeiT-B ⊙M$_{VIOLIN}$ | 81.33 | 61.31 | 53.93 | 83.22 | 81.72 | 57.28 | 35.37 | 40.98 |
| | DeiT-B LoRA | 79.1 | 60.15 | 51.93 | 81.25 | 78.53 | 53.71 | 28.28 | 32.12 |
| | DeiT-B ⊙M$_{VIOLIN}$ LoRA | 82.36 | 63.46 | 52.86 | 82.18 | 78.52 | 55.25 | 32.21 | 39.79 |
| | DeiT-B DoRA | 76.97 | 60.62 | 50.37 | 81.34 | 73.34 | 54.11 | 28.69 | 39.43 |
| | DeiT-B ⊙M$_{VIOLIN}$ DoRA | 81.64 | 63.29 | 51.06 | 82.42 | 78.65 | 56.14 | 27.62 | 38.89 |
| | DeiT-III-S | 76.53 | 57.29 | 46.23 | 81.81 | 58.12 | 50.48 | 26.33 | 26.57 |
| | DeiT-III-S ⊙M$_{VIOLIN}$ | 77.78 | 61.9 | 54.84 | 83.17 | 85.91 | 59.78 | 33.45 | 36.07 |
| | DeiT-III-B | 80.54 | 61.82 | 50.95 | 82.7 | 60.75 | 55.35 | 30.36 | 31.18 |
| | DeiT-III-B ⊙M$_{VIOLIN}$ | 84.51 | 61.92 | 55.64 | 82.79 | 84.06 | 60.34 | 36.59 | 38.4 |
| | DeiT-III-L | 72.99 | 53.23 | 47.59 | 80.78 | 50.19 | 50.72 | 25.21 | 30.51 |
| | DeiT-III-L ⊙M$_{VIOLIN}$ | 76.66 | 55.64 | 50.03 | 81.86 | 55.42 | 57.35 | 28.69 | 33.91 |
| | DeiT-III-H | 75.17 | 55.24 | 48.66 | 81.11 | 41.57 | 46.99 | 25.15 | 31.74 |
| | DeiT-III-H ⊙M$_{VIOLIN}$ | 77.89 | 55.96 | 50.96 | 81.9 | 47.85 | 55.07 | 26.57 | 33 |
| | DINO-S | 83.29 | 65.03 | 53.44 | 80.03 | 78.72 | 48.61 | 34.23 | 41.87 |
| | VIOLIN DINO-S | 84.19 | 63.35 | 55.72 | 81.43 | 75.82 | 49.37 | 32.92 | 42.06 |
| | DINO-S ⊙M$_{VIOLIN}$ | 83.69 | 64.23 | 55.35 | 79.98 | 79.42 | 49.18 | 36.43 | 41.61 |
| | DINO-B | 80.93 | 62.76 | 52.17 | 79.23 | 69.22 | 48.39 | 33.73 | 41.34 |
| | VIOLIN DINO-B | 81.96 | 63.04 | 53.45 | 79 | 72.12 | 49.59 | 30.29 | 40.76 |
| | DINO-B ⊙M$_{VIOLIN}$ | 83.87 | 63.65 | 55.66 | 81.2 | 74.14 | 54.18 | 34.79 | 39.27 |

Table 21: *VTAB Results-Specialized Subset:* Individual scores for each dataset. SN refers to Small-Norm, and dS represents dSprites.

| | Model | Patch Camelyon | EuroSAT | Resisc45 | Diabetic Retinopathy |
|---|---|---|---|---|---|
| **Specialized** | DeiT-T | 82.79 | 93.53 | 80.98 | 72.05 |
| | VIOLIN DeiT-T | 82.47 | 93.35 | 81.3 | 73.43 |
| | DeiT-T ⊙M$_{VIOLIN}$ | 84.04 | 93.88 | 83.23 | 73.87 |
| | DeiT-S | 84.08 | 94.4 | 84.01 | 74.72 |
| | VIOLIN DeiT-S | 85.36 | 95.41 | 83.86 | 74.85 |
| | DeiT-S ⊙M$_{VIOLIN}$ | 85.19 | 95.02 | 85.68 | 74.32 |
| | DeiT-B | 85.74 | 95.38 | 86.37 | 74.6 |
| | VIOLIN DeiT-B | 85.62 | 95.44 | 85.68 | 75.02 |
| | DeiT-B ⊙M$_{VIOLIN}$ | 86.74 | 95.91 | 87.31 | 75.2 |
| | DeiT-B LoRA | 86.2 | 95.46 | 85.72 | 75.09 |
| | DeiT-B ⊙M$_{VIOLIN}$ LoRA | 85.9 | 95.66 | 86.71 | 73.73 |
| | DeiT-B DoRA | 85.53 | 95.39 | 85.21 | 74.8 |
| | DeiT-B ⊙M$_{VIOLIN}$ DoRA | 85.92 | 95.56 | 84.98 | 73.35 |
| | DeiT-III-S | 84.57 | 93.33 | 82.68 | 73.94 |
| | DeiT-III-S ⊙M$_{VIOLIN}$ | 85.76 | 94.98 | 86.43 | 74.67 |
| | DeiT-III-B | 86.4 | 94.47 | 85.83 | 74.33 |
| | DeiT-III-B ⊙M$_{VIOLIN}$ | 87.77 | 95.8 | 87.57 | 74.73 |
| | DeiT-III-L | 84.5 | 93.28 | 84.47 | 75.28 |
| | DeiT-III-L ⊙M$_{VIOLIN}$ | 84.54 | 94.11 | 85.24 | 74.83 |
| | DeiT-III-H | 84.64 | 92.64 | 84.99 | 74.46 |
| | DeiT-III-H ⊙M$_{VIOLIN}$ | 84.81 | 93.3 | 84.66 | 74.93 |
| | DINO-S | 86.82 | 94.29 | 86.13 | 73.14 |
| | VIOLIN DINO-S | 87.7 | 94.76 | 86.59 | 73.96 |
| | DINO-S ⊙M$_{VIOLIN}$ | 85.94 | 94.9 | 86.17 | 74.26 |
| | DINO-B | 87.02 | 94.45 | 87.05 | 74.55 |
| | VIOLIN DINO-B | 87.57 | 94.46 | 87.25 | 74.03 |
| | DINO-B ⊙M$_{VIOLIN}$ | 87.81 | 95.44 | 87.96 | 74.54 |

## F.6 COMPARISON AGAINST OTHER LOCALITY METHODS.

There are many methods for enhancing locality in plain ViTs. To compare these approaches with VIOLIN , we start from the same pretrained DeiT-B model, add each locality mechanism on top of it, and fine-tune all models under the exact same protocol. This ensures that every method begins from an identical initialization. The results show that while all methods offer some improvement, VIOLIN achieves the strongest gains. Below, we detail how each method is incorporated and initialized to preserve the pretrained model at the start of fine-tuning, and we report results in Tables 22 to 24.

**Swin RPB**   Swin transformers (Liu et al., 2021) introduces locality two ways, by partitioning the feature map into shifted windows, and with relative position biases (RPB) that encode spatial offsets inside each window. These biases give the attention mechanism information about relative spatial relationships within a window, improving performance on vision tasks where nearby pixels are correlated. To incorporate RPB into a pretrained global-attention ViT, we add a learnable bias term $\mathbf{B} \in \mathbb{R}^{N \times N}$ as in eq. (49) where $\mathbf{B}[i, j]$ depends on the relative position of the tokens $i$ and $j$.

$$\mathbf{Y} = \text{Softmax}\left(\frac{\mathbf{Q}\mathbf{K}^\top}{\sqrt{d}} + \mathbf{B}\right)\mathbf{V}. \tag{49}$$

By initializing $\mathbf{B}$ with zeros, the modified attention reduces exactly to the original attention. This guarantees that adding the Swin-RPB does not alter the models capabilities and new positional biases can be learned during fine-tuning.

**2D Relative Position Encoding (iRPE)**   iRPE (Wu et al., 2021a) add locality into attention, by adding learnable bias terms based on the 2-D relative position of tokens. For any pair of tokens $(i, j)$, the offset $\Delta p_{ij}$ is mapped through a bucketing function to an index $b_{ij}$, which selects a bias embedding from a table $R \in \mathbb{R}^{B \times H}$. Depending on the chosen attachment mode, this embedding is added to queries, keys or values (e.g., $\hat{k}_j = k_j + R_{b_{\cdot j}}$) and the attention scores are calculated using this new parameters. To integrate iRPE into a pretrained ViT without disturbing its learned representations, we initialize all bucket embeddings to zero,

$$R_b = 0 \;\; \forall\, b$$

so that the queries/keys/values are not changed at the start of finetuning. This ensures that the model initially behaves exactly like the pretrained backbone, while the RPE parameters gradually learn non-zero spatial biases during training.

**LocalVit**   LocalVit (Li et al., 2021) enhances locality inside the feed-forward network (FFN) rather than attention. It replaces the MLP with a depthwise-convolutional residual branch. This allows each token to mix information with its spatial neighbors, giving the transformer an inductive bias similar to CNNs while preserving the global interactions of self-attention. For LocalViT, we gate the convolutional branch with a learnable scalar initialized to zero, and initialize the depthwise conv as an identity kernel (center=1, others=0). This allows the modified architecture to behave exactly the same as the pretrained model at the first step, enabling smooth fine-tuning and gradual learning of locality information.

**VIOLIN variations**   Additionally, we evaluate several ablations discussed in previous sections, including an additive version of MVIOLIN , Manhattan-distance masking, a single-curve variant (MPeano), and random-curve masking (M_{Random}), under the same finetuning protocol for completeness. These results further highlight the contributions of using multiple SFCs rather than relying on any single locality pattern.

Table 22: *VTAB Results-Natural Subset:* Individual scores for each dataset for different locality-enforcing methods.

| Model | CIFAR | Caltech101 | DTD | Flowers102 | Pets | SVHN | Sun397 |
|---|---|---|---|---|---|---|---|
| Additive $M_{\text{VIOLIN}}$ | 63.64 | 91.11 | 69.27 | 93.6 | 92.6 | 90.46 | 44.28 |
| Swin RPB | 63.72 | 90.75 | 70.16 | 94.15 | 92.66 | 90.21 | 45.82 |
| i-RPE-QKV | 65.03 | 90.94 | 70.12 | 93.97 | 92.63 | 90.32 | 45.66 |
| LocalVit | 65.17 | 91.13 | 69.57 | 93.85 | 92.56 | 90.26 | 45.63 |
| Manhattan | 59.62 | 90.78 | 68.03 | 92.07 | 91.47 | 89.81 | 42.13 |
| $M_{\text{Peano}}$ | 65.04 | 90.78 | 69.18 | 94.11 | 92.61 | 90.14 | 45.89 |
| $M_{\text{Random}}$ | 65.02 | 90.78 | 69.02 | 94.09 | 92.6 | 89.74 | 45.91 |

Table 23: *VTAB Results-Structured Subset:* Individual scores for each dataset for different locality-enforcing methods. SN refers to SmallNorm, and dS represents dSprites.

| Model | CLEVR Count | CLEVR Dist | DMLab | KITTI | dS Loc | dS Ori | SN Azi | SN Ere |
|---|---|---|---|---|---|---|---|---|
| Additive $M_{\text{VIOLIN}}$ | 81.08 | 62.12 | 51.95 | 83.26 | 80.95 | 57.25 | 34.76 | 39.38 |
| Swin RPB | 81.42 | 61.67 | 53.83 | 83.17 | 81.39 | 56.81 | 35.42 | 38.9 |
| i-RPE-QKV | 81.25 | 61.58 | 53.42 | 83.12 | 81.49 | 57.28 | 35.13 | 38.34 |
| LocalVit | 81.28 | 61.53 | 53.43 | 82.56 | 81.38 | 57.6 | 35.5 | 38.71 |
| Manhattan | 76.74 | 60.73 | 50.16 | 82.51 | 74.69 | 55.03 | 32.49 | 34.57 |
| $M_{\text{Peano}}$ | 81.45 | 61.4 | 53.59 | 83.17 | 81.09 | 56.98 | 34.53 | 40.84 |
| $M_{\text{Random}}$ | 81.45 | 61.33 | 53.36 | 82.84 | 80.21 | 56.98 | 34.5 | 40.76 |

Table 24: *VTAB Results-Specialized Subset:* Individual scores for each dataset for different locality-enforcing methods.

| Model | Patch Camelyon | EuroSAT | Resisc45 | Diabetic Retinopathy |
|---|---|---|---|---|
| Additive $M_{\text{VIOLIN}}$ | 86.84 | 96.07 | 87.62 | 74.93 |
| Swin RPB | 86.17 | 95.66 | 87.47 | 75.37 |
| i-RPE-QKV | 86.76 | 95.72 | 87.51 | 74.91 |
| LocalVit | 86.55 | 95.85 | 87.58 | 75.4 |
| Manhattan | 86.44 | 94.93 | 86.29 | 74.21 |
| $M_{\text{Peano}}$ | 87.13 | 95.93 | 87.71 | 75.56 |
| $M_{\text{Random}}$ | 86.8 | 95.57 | 87.63 | 75.26 |

## F.7 LEARNED CURVE ORDER

Motivated by recent work on learned patch orderings (Kutscher et al., 2025), we implemented a learned ordering variant within our framework and trained a DeiT-Tiny model using this learned sequence. The results are shown in Table Table 25. Although the learned variant underperforms the original VIOLIN mask in this initial experiment, it highlights several promising research directions, such as jointly learning multiple traversal curves, exploring task-adaptive orderings, and studying how different datasets induce specialized spatial structuresall of which may further improve performance and interpretability.

Table 25: *Comparison of DeiT-Tiny, VIOLIN, and a learned patch-ordering variant:* learned patch orderings (Kutscher et al., 2025) is adapted to VIOLIN framework.

| Model | Accuracy (%) |
|---|---|
| DeiT-Tiny | 72.2 |
| VIOLIN | 73.0 |
| VIOLIN w learned order | 70.1 |

### F.8 Comparison with relative positional encodings

VIOLIN and relative positional encodings (RPEs) introduce spatial inductive bias through different mechanisms. As described in Appendix B.4, VIOLIN applies a lightweight multiplicative decay mask, whereas modern RPEs add learned pairwise positional terms to the attention logits and often require additional parameters or architecture-specific modifications. To assess their relationship, in addition to the fine-tuning experiments in Appendix F.6, we include comparisons with several RPE-based locality baselines in both the pretraining settings.

On ImageNet-1K supervised pretraining, VIOLIN achieves competitive performance to several RPE variants while adding significantly fewer FLOPs. For example, on DeiT-S, VIOLIN introduces $5\times$ fewer FLOPs than Transformer-XL and $1.3\times$ fewer FLOPs than iRPE-QK, while obtaining comparable accuracy.

Table 26: *Comparison of* VIOLIN *and RPE variants:* on DeiT-S pretraining in ImageNet-1K. Results are taken from respective papers of i-RPE (Wu et al., 2021a) and Transformer-XL Dai et al. (2019).

| Model | Additional FLOPs (%) | Top-1 Acc. (%) |
|---|---|---|
| Baseline | - | 79.9 |
| VIOLIN | 0.7 | 80.7 |
| Transformer-XL | 4.3 | 80.8 |
| iRPE-K | 0.9 | 80.9 |
| iRPE-QK | 2.2 | 81.1 |
| iRPE-QKV | 5.9 | 81.4 |

VIOLIN can also be combined with RPEs. On DeiT-T, adding VIOLIN to iRPE-K yields an additional accuracy gain, indicating that the methods introduce complementary inductive information.

Table 27: *Combination of* VIOLIN *with RPEs:* pretraining results on DeiT-T model as baseline, with PRE and with RPE+VIOLIN .

| Model | Additional FLOPs (%) | Top-1 Acc. (%) |
|---|---|---|
| Baseline | – | 72.2 |
| iRPE-K | 1.7 | 73.7 |
| iRPE-K + VIOLIN | 2.3 | 73.9 |

# G  CODES AND IMPLEMENTATION DETAILS

## G.1  COMPUTE RESOURCES

Table 28: *Compute resources for pertaining:* The number of GPUS and approximate training time for each model and scale are provided.

| Model | # GPUs | Training time |
|-------|--------|---------------|
| DeiT-T | 4 | $\approx$ 17 Hour |
| DeiT-S | 4 | $\approx$ 23 Hour |
| DeiT-B | 16 | $\approx$ 1.7 Day |
| DINO-S | 16 | $\approx$ 3.2 Days |
| DINO-B | 16 | $\approx$ 7 Days |

In Table 28, we report the compute resources required for each of the evaluated models. These numbers also apply to the models used for ablation experiments.

For fine-tuning, we performed 30 runs per dataset for each model (25 for validation and 5 for final evaluation). Each run took between 2 to 10 minutes, and the complete fine-tuning evaluation was completed in approximately 10 days.

All experiments were conducted using a mix of NVIDIA A100 SXM4 80GB, NVIDIA GH200 96GB, and NVIDIA H100 SXM5 80GB GPUs, used interchangeably depending on availability.

## G.2  VTAB-1K HYPERPARAMETERS

To determine optimal learning rates, we use the VTAB-1K-pytorch repository (Alkin, 2022) and conduct a grid search. Following the original implementation, we run 5 seeds for learning rate selection on validation set and another 5 seeds for standard training. For each model, we average the top 3 runs to report the final accuracy. The complete list of hyperparameters is provided in Table 29. For parameter-efficient fine-tuning, we again use the same set of hyperparameters and grid search over ranks [2,4,8,16].

Table 29: *Hyperparameters for fine-tuning on VTAB-1K*: The same hyperparameters are used for all models, following (Alkin, 2022).

| Parameter | Value |
|-----------|-------|
| Epochs | 50 |
| Batch size | 64 |
| Seeds | 5 |
| Optimizer | AdamW |
|    Learning rate | [1e-3, 7.5e-4, 5.0e-4, 2.5e-4, 1.0e-4] |
|    Layer-wise lr deca | 0.65* |
|    Weight decay | 0.05 |
|    Momentum | $\beta_1 = 0.9, \beta_2 = 0.999$ |
| Learning rate schedule | linear warmup $\rightarrow$ cosine decay |
|    Warmup epochs | 5 |
| Precision | mixed `bfloat16` |
|    Backend | `torch.autocast` |
| Data Augmentation | |
|    `Resize` | |
|      `interpolation` | bicubic |
|      `size` | 224x224 |
|    `Normalize` | ImageNet-1K statistics |

## G.3 CODES FOR CURVES

In this section, we provide the codes used to create the permutation orders of each SFC in basis of Z-curve. In other words, we define efficiency the indexing needed for the permutation $\pi_c(.)$ for each curve $c$ used in our study.

**Snake curve**

```python
def snake_curve(grid):
    """Returns the elements of the grid in snake order."""
     n_rows, n_cols = grid.shape
    order = []
    for y in range(n_rows):
        if y % 2 == 0:
            # Left-to-right for even rows
            order.extend((x, y) for x in range(n_cols))
        else:
            # Right-to-left for odd rows
            order.extend((x, y) for x in reversed(range(n_cols)))
    return order
```

**Zig-zag curve**

```python
def zigzag_curve(grid):
    """Returns the elements of the grid in diagonal zig-zag order."""
    n_rows, n_cols = grid.shape
    order = []
    for d in range(n_rows + n_cols - 1):
        if d % 2 == 0:
            r = min(d, n_rows - 1)
            c = d - r
            while r >= 0 and c < n_cols:
                order.append((r, c))
                r -= 1
                c += 1
        else:
            c = min(d, n_cols - 1)
            r = d - c
            while c >= 0 and r < n_rows:
                order.append((r, c))
                c -= 1
                r += 1
    return order
```

**Hilbert curve** Adapted from (Cerveny, 2025).

```python
def hilbert_curve(grid):
    rows = len(grid)
    cols = len(grid[0]) if rows > 0 else 0
    return [(x, y) for x,y in gilbert2d(rows, cols)]

def gilbert2d(width, height):
    """
    Generalized Hilbert ('gilbert') space-filling curve for arbitrary
                                   -sized
    2D rectangular grids. Generates discrete 2D coordinates to fill a
                                   rectangle
    of size (width x height).
    """
    if width >= height:
        yield from generate2d(0, 0, width, 0, 0, height)
    else:
        yield from generate2d(0, 0, 0, height, width, 0)

def sgn(x):
    return -1 if x < 0 else (1 if x > 0 else 0)

def generate2d(x, y, ax, ay, bx, by):
    w = abs(ax + ay)
    h = abs(bx + by)
    (dax, day) = (sgn(ax), sgn(ay)) # unit major direction
    (dbx, dby) = (sgn(bx), sgn(by)) # unit orthogonal direction
    if h == 1:
        # trivial row fill
        for i in range(0, w):
            yield(x, y)
            (x, y) = (x + dax, y + day)
        return
    if w == 1:
        # trivial column fill
        for i in range(0, h):
            yield(x, y)
            (x, y) = (x + dbx, y + dby)
        return
    (ax2, ay2) = (ax//2, ay//2)
    (bx2, by2) = (bx//2, by//2)
    w2 = abs(ax2 + ay2)
    h2 = abs(bx2 + by2)
    if 2*w > 3*h:
        if (w2 % 2) and (w > 2):
            # prefer even steps
            (ax2, ay2) = (ax2 + dax, ay2 + day)
        # long case: split in two parts only
        yield from generate2d(x, y, ax2, ay2, bx, by)
        yield from generate2d(x+ax2, y+ay2, ax-ax2, ay-ay2, bx, by)
    else:
        if (h2 % 2) and (h > 2):
            # prefer even steps
            (bx2, by2) = (bx2 + dbx, by2 + dby)
        # standard case: one step up, one long horizontal, one step
                                           down
        yield from generate2d(x, y, bx2, by2, ax2, ay2)
        yield from generate2d(x+bx2, y+by2, ax, ay, bx-bx2, by-by2)
        yield from generate2d(x+(ax-dax)+(bx2-dbx), y+(ay-day)+(by2-
                                           dby),
                                   -bx2, -by2, -(ax-ax2), -(ay-ay2))
```

**Peano curve** Adapted from (Schubotz, 2021; Prater).

```python
def interleave_bits(x, y):
    """
    Interleave the bits of two integers (x, y) to compute Morton
                                         order.
    """
    def split_bits(value):
        result = 0
        for i in range(32):  # Support up to 32-bit integers
            result |= ((value >> i) & 1) << (2 * i)
        return result

    return split_bits(x) | (split_bits(y) << 1)

def peano_curve(grid):
    """Returns the elements of the grid in diagonal morton/peano
                                       order."""
    n_rows, n_cols = grid.shape
    order = []

    for y in range(n_rows):
        for x in range(n_cols):
            morton_key = interleave_bits(x, y)
            order.append((morton_key, x, y))

    # Sort by Morton key to achieve the Morton curve order
    order.sort(key=lambda pair: pair[0])
    return [(x, y) for _, x, y in order]
```

### G.4 CODE OF EFFICIENT DECAY MASK

```python
def Casual_Decay_Mask(b_i , N):
    idx = torch.arange(N,device=b_i.device)
    I, J = torch.meshgrid(idx, idx, indexing='ij')
    E = (torch.abs((I-J)).float().view(1,1,N,N))
    M = torch.sigmoid(b_i).view(1,-1,1,1)**E
    return M
```

### G.5 THE USE OF LARGE LANGUAGE MODELS (LLMS)

While preparing this manuscript, we limitedly used Large Language Models (LLMs). Their role was restricted to assisting with editing and polishing the writing, such as improving clarity, grammar, and flow. All conceptual ideas, methods, experiments, and analyses presented in this paper are entirely the work of the authors. No ideas, algorithms, or research contributions were generated by an LLM. The models served only as a tool to refine the presentation of the text without influencing the substance of the research.

