# OpenReview forum: "Space Filling Curves as Spatial Priors for Small or Data-Scarce Vision Transformers"
_ICLR.cc/2026/Conference — Submitted to ICLR 2026_

### Official Review · Reviewer_sQfG · 2025-10-30

**Soundness:** 2
**Presentation:** 2
**Contribution:** 2
**Rating:** 4
**Confidence:** 3

**Summary:**

This paper proposes **VIOLIN**, a simple and plug-and-play spatial prior module for Vision Transformers (ViTs).
The method introduces *Space Filling Curves (SFCs)* (e.g., Snake, Zig-zag, Peano, Hilbert) to define alternative scanning orders of image patches.
For each curve \(c\), a decaying mask \(M_c[i,j] = \gamma_c^{|i-j|}\) is constructed to encourage locality in attention.
After aligning these masks back to the standard patch order and averaging, the resulting mask \(M_{\text{VIOLIN}}\) is multiplied with the attention score matrix before softmax.

The approach is extremely lightweight (+0.0002% params, +0.64% FLOPs) and can be applied to pretrained or finetuned ViTs without architectural changes.
Extensive experiments on **VTAB-1K**, **ImageNet-1K**, **DINO**, **pixel-level CIFAR-100**, and dense tasks (ADE20K / COCO) show consistent gains, especially on “Structured” VTAB tasks (+8.7%).

**Strengths:**

- **Well-defined target problem:** Focuses on *small models and data-scarce regimes* where ViTs lack spatial inductive bias — a meaningful and under-explored setting.
- **Simplicity and generality:** VIOLIN requires no retraining or re-architecture changes, making it truly plug-and-play.
- **Elegant formulation:** The SFC-based decaying masks are clearly derived; the permutation and averaging operations are well explained.
- **Strong empirical results:** Significant improvement on VTAB-1K (Structured group +8.7%) and pixel-level CIFAR-100 (+7.2%) convincingly show the benefit of spatial priors.
- **Low computational cost:** The added overhead is negligible, suitable for real-world low-resource finetuning.
- **Broad applicability:** Small but consistent gains on segmentation and detection tasks further validate its generality.

**Weaknesses:**

1. **Novelty is limited.**
   The core idea—distance-decayed attention weights—is reminiscent of *linear attention*, *RMT*, and *RetNet*–style exponential decay mechanisms.
   The use of multiple SFCs and their averaged mask is an incremental extension rather than a fundamentally new concept.

2. **Missing comparisons with strong baselines.**
   The paper compares mainly to vanilla DeiT/DeiT-III/DINO backbones.
   It lacks direct comparisons with existing locality-enforcing methods, such as:
   - Relative positional bias (Swin / ViT-RPB),
   - Convolutional stems or LocalViT,
   - Manhattan-distance masks (RMT),
   - Single-curve or random-curve baselines.
   Without these, it is unclear whether the large Structured-task gains stem from the proposed multi-SFC averaging or from any reasonable local bias.

3. **Training details and fairness are under-specified.**
   VTAB-1K finetuning recipes (learning rate, γ initialization, α sharing) are buried in the appendix.
   It remains unclear whether baselines were tuned equivalently.
   The surprising claim that *untrained masks outperform pretrained ones* needs stronger justification.

4. **Questionable mask effectiveness.**
   Figure 7 shows most γ₍c₎ values approach 1, suggesting the mask becomes nearly uniform.
   If so, why does the Structured group improve so dramatically?
   More analysis of per-head γ values and locality visualization is needed.

5. **Computational overhead claim is not empirically verified.**
   Only theoretical FLOPs/parameter ratios are reported.
   Actual GPU memory and runtime increase (especially on dense tasks) should be measured.

6. **Overstated framing.**
   The paper sometimes overclaims by calling VIOLIN a *principled spatial prior via SFCs*.
   In fact, the method does not exploit the geometric guarantees of SFCs; it only uses index distance \(|i-j|\) with exponential decay.
   Theoretical justification for averaging multiple SFC-induced metrics is weak.

**Questions:**

Please refer to Weaknesses

---

> ### Author Response · Authors · 2025-11-19
>
> We thank the reviewer for the careful reading of our work and for highlighting the **clarity, simplicity, and empirical strengths** of VIOLIN. We appreciate the recognition of our motivation and the method’s **broad applicability**. Below we address the raised concerns.
>
> ---
> ### **1. Limited novelty**
>
> While decay-based attention mechanisms exist (e.g., linear attention, RMT, RetNet), VIOLIN is orthogonal to these methods. Our contribution is **not a new attention formulation but a new way to inject spatial inductive bias into standard ViTs**. This differs from prior work in several key ways:
>
> * It uses multiple structured SFCs to construct attention decay masks, this is, to our knowledge, new in Vision Transformers.
> * Averaging the learned decay masks from several SFCs provides a richer and more flexible spatial prior than any single distance metric.
> * Introduces an efficient Z-curve basis alignment, allowing all SFC masks to share the same $QK^T$ computation and making multi-SFC priors practical and lightweight.
>
> As a result, VIOLIN presents a novel solution for the problem of injecting spatial inductive bias into ViTs, especially in small-model, low-data regimes. Moreover, VIOLIN is highly effective even when applied only at finetuning, other approaches generally lack this flexibility.
>
> ---
> ### **2. Missing comparisons**
>
> We appreciate the suggestion. In Appendix E.2, we already include **Manhattan-distance, single-curve**, and **random-curve** baselines in the pretraining setting. Following the reviewer’s comment, we additionally tested all the suggested locality-enforcing methods in the finetuning setting.
>
> For each approach, we used the **same pretrained DeiT-B backbone** and properly initialized the corresponding locality mechanism on top of it, ensuring that all models start from an identical initialization. We then finetuned all methods under the same protocol, using the hyperparameter search described in Appendix G.2. In the table below, we present the results on the Structured-group.
>
> | **Method** | ** Extra Parameters** |**Structured Avg. (%)** |
> |---|---|---|
> | Baseline (DeiT-B) |-| 57.00 |
> | **VIOLIN (ours)** | ~1.3K  |**61.89**|
> | Additive VIOLIN | ~1.3K |61.34|
> | Swin RPB | ~105K|61.58|
> | i-RPE QKV| ~115K|61.45 |
> | LocalViT | ~6.2M|61.50 |
> | Manhattan Mask | ~0.4K  | 58.37 |
> | Single SFC (Peano) | ~0.4K |61.63 |
> | Random Curve | ~0.4K |61.43 |
>
> Results indicate that most locality priors offer some improvement, but VIOLIN provides **the strongest gains with minimal overhead**, showing that the improvements come specifically from the multi-SFC averaging, not just any local bias.
>
> All implementation details, initialization procedures, and per-dataset results have been added to Section 4.3 and Appendix F.6 (highlighted in blue).
>
> ---
> ### **3. Training details**
>
> All methods, both our approach and all baselines, were tuned under the exact same hyperparameter search, random seeds, and training protocol described in Appendix G.2. We added clearer explanations in the main text to make this more explicit.
>
> Thanks to your feedback, we recognized that the “untrained mask” claim was unclear and the original wording was confusing. To clarify, the finetuning procedure used to for the results in Table 3 is as follows:
>
> * We start from a **pretrained vanilla backbone** (e.g., DeiT).
> * We then add a **freshly initialized VIOLIN mask** on top of this backbone.
> * During finetuning, **both the backbone and the mask are trained jointly**.
>
> In contrast, the Appendix F.2 experiment starts from **pretrained VIOLIN models**, i.e., models that were already pretrained with the VIOLIN mask active, and then fine-tuned under the same finetuning setup.
>
> The first setting (with freshly initialized mask) offers the model **more flexibility to learn task-specific spatial priors during fine-tuning**, which explains why they perform better. We have updated the manuscript to clearly reflect this distinction and revised the language to avoid any ambiguity.
>
> ---
> ### **4. Mask effectiveness**
>
> Thanks for the sharp question. Indeed, $\gamma \sim 1$ is common for linear transformers and state-space models [1]. Specifically, in LRU [2], it is shown that when $\gamma \sim 1$, the model can unlock long-range reasoning capabilities, and during training most of the $\gamma$ values converges towards 1.
>
> With a squence of length 196, even moderate decay (e.g., 0.9^196 < 10⁻⁹) drastically reduces the receptive field. Values close to 1 are therefore standard and help to preserve **both local and global information**.
>
> In this context, $\gamma\sim1$ in VIOLIN, does **not indicate an ineffective mask**. Larger $\gamma$ values simply produce broader receptive fields, while smaller values (e.g., $\gamma < 0.5$) result in more local attention patterns. We have added additional mask visualizations with different $\gamma$ values in Appendix F.4.

---

> > ### Author Response · Authors · 2025-11-19
> >
> > ### **5. Computational overhead**
> >
> > Thank you for the suggestion. We have now measured the actual average inference GPU memory usage and runtime for a batch size of 256 at two resolutions (224×224 for classification and 512×512 for dense tasks), comparing vanilla DeiT and VIOLIN on the same hardware. The results closely match our theoretical analysis and confirm that VIOLIN introduces **only minimal overhead in both memory and runtime**. We have added these measurements to the manuscript for completeness.
> >
> > | **Model** | **GPU Memory (GB)** |**Run time (ms/batch)** |
> > |---|---|---|
> > |DeiT (res. 224)|0.80| 206.1
> > |VIOLIN (res. 224)|0.81|233.1|
> > |DeiT (res. 512)|13.88|1739.3|
> > |VIOLIN (res. 512)|13.90| 1789.7|
> >
> > ---
> > ### **6. Overstated framing**
> >
> > Thank you for the comment. We have revised our wording to better reflect the claims of our paper, specifically by rephrasing the statement “a principled spatial prior via SFCs.”
> >
> > ---
> > ### **7. Theoretical justification for averaging multiple SFC**
> >
> > VIOLIN operates on a single $QK^{\top}$ matrix to avoid creating seperate attentions for each curve and maintain efficiency. This requires merging all SFC-specific masks into a single mask $M$ that aligns with the ordering of $QK^{\top}$, and thus any multi-curve integration necessarily involves an element-wise operation of the form $M \odot QK^{\top}$. In this setting, averaging is a natural choice: it is stable, incurs no runtime overhead, and retains information from all curves. We also explored alternatives such as learned weighting and gating, but these increased complexity and parameter count without improving performance.
> >
> > ---
> >
> > [1] Y. Sun et al., . Retentive Network: A Successor to Transformer for Large Language Models, 2023.
> >
> > [2] A. Orvieto et al., Resurrecting Recurrent Neural Networks for Long Sequences, 2023.

---

### Official Review · Reviewer_nX45 · 2025-10-31

**Soundness:** 2
**Presentation:** 2
**Contribution:** 1
**Rating:** 2
**Confidence:** 4

**Summary:**

The paper proposes VIOLIN, a masked attention mechanism for Vision Transformers (ViTs) that incorporates Space Filling Curves (SFCs) to improve spatial inductive biases. Standard ViTs suffer from the lack of spatial awareness due to the permutation-equivalent nature of self-attention. Inspired by linear attention and SSMs, VIOLIN constructs curve-specific decay masks that model the relative spatial distance between image patches. These masks are averaged and applied to the attention matrix, introducing spatial priors without modifying the core ViT architecture.

**Strengths:**

- Extensive empirical validation: tested on diverse model scales (5M–86M parameters) and training setups (supervised and self-supervised).

- The paper is easy to follow.

- The proposed method shows some improvment.

**Weaknesses:**

- **Limited Contribution from the Core Method**: [6] has shown that average pooling can boost the DeiT's performance. Tab. 14 suggests that **the performance gain mainly comes from the average pooling**. The VIOLIN only provide marginal improvement for small models, and **even harms the performance of the large model ViT-B**.

- **Limited Generalization**: Based on the the results in Tab. 8, **the improvements on Swin-T and Swin-S are below 0.2%**, which is likely within run-to-run variance and not statistically significant. This suggests that the proposed method  is rather an engineering optimization technique, which does not generalize well to different models.

- **Limited Comparison**. The baselines used for comparison primarily rely on absolute positional embeddings, which are known to be suboptimal. Relative positional encodings, widely adopted in modern architectures [1-5], are simpler, more flexible, and have been shown to outperform absolute encodings in multiple settings. **It is not clear that space-filling curves offer any meaningful advantage over such approaches**. Without direct comparisons to relative positional encoding, the benefits of VIOLIN are difficult to justify.

[1] Wu, Kan, et al. "Rethinking and improving relative position encoding for vision transformer." Proceedings of the IEEE/CVF international conference on computer vision. 2021.

[2] Zihang Dai, Zhilin Yang, Yiming Yang, Jaime G Carbonell, Quoc Le, and Ruslan Salakhutdinov. Transformer-xl: Attentive language models beyond a fixed-length context. In ACL, 2019. 1, 3, 7, 8

[3] Liu, Ze, et al. "Swin transformer: Hierarchical vision transformer using shifted windows." Proceedings of the IEEE/CVF international conference on computer vision. 2021.

[4] Liu, Ze, et al. "Swin transformer v2: Scaling up capacity and resolution." Proceedings of the IEEE/CVF conference on computer vision and pattern recognition. 2022.

[5] Zhou, Yuxuan, et al. "SP-ViT: Learning 2D Spatial Priors for Vision Transformers." 33rd British Machine Vision Conference. BMVA Press, 2022.

[6] Conditional Positional Encodings for Vision Transformers, ICLR2023.

**Questions:**

Could the authors also compare VIOLIN to other methods related to spatial prior, such as relative positional encoding/bias in [1-5] ?

---

> ### Author Response · Authors · 2025-11-26
>
> We thank the reviewer for the assessment of our work and for noting the clarity of the presentation, **the breadth of the empirical validation across model sizes and training regimes**, and **the improvements** brought by our approach. Below, we address the reviewer’s specific concerns.
>
> ---
> ### **1. Limited Contribution**
>
> This claim is incorrect based on the results presented in the current manuscript. Appendix E.5 and Table 16 (Table 14 pre-rebuttal) explicitly distinguish the effect of GAP from the effect of VIOLIN. As shown in the table below, GAP and VIOLIN contribute separate and comparable improvements:
>
> * GAP alone yields +0.4 performance increase in both tiny and small scale.
>
> * VIOLIN provides a **further improvement** of **+0.4** (DeiT-T), **+0.5** (DeiT-S), on top of GAP.
>
> These gains **are not marginal**, and the numbers clearly show that VIOLIN does not derive its benefit from GAP. This is also discussed in Appendix E.5, where we explain why VIOLIN’s spatial bias and GAP complement one another.
>
> Regarding the comment on DeiT-B, the paper is **explicitly targeted to the small-model, small-data regime**, as stated throughout the title, abstract, and main text. Consequently, the base-scale model is not the primary target, our focus is on DeiT-T and DeiT-S, where VIOLIN delivers the most meaningful gains.
>
> | **Model** | **CLS Baseline** | **CLS + VIOLIN** | **GAP Baseline** | **GAP + VIOLIN** |
> |----------|-------------------|-------------------|--------------------|-------------------|
> | DeiT-T | 72.2 | 72.3 (+0.2)| 72.6 | **73.0** (+0.8) |
> | DeiT-S | 79.8 | 80.1 (+0.3)| 80.2 | **80.7** (+0.9)|
>
>
> ---
> ### **2. Limited Generalization**
>
> As discussed in **Appendix D.2**, VIOLIN is intended for architectures that use **global attention**, where a global SFC ordering can meaningfully shape the receptive field. Swin relies on local windowed attention, which already provides a strong built-in spatial inductive bias and restricts interactions to small neighborhoods. In such settings, any additional global spatial prior naturally has limited influence.
>
> Given this architectural mismatch, the small improvements observed on Swin-T and Swin-S are **expected and consistent with our analysis**. Rather than indicating weak generalization, these results simply reflect that Swin already enforces locality structurally, leaving little room for an external prior to add further benefit.
>
> Importantly, the scope of our work is the small-model, global-attention ViT regime, where spatial priors are most impactful. In this intended setting, VIOLIN generalizes well and consistently yields substantial gains in various models and training settings.

---

> > ### Author Response · Authors · 2025-11-26
> >
> > ### **3. Limited Comparison**
> >
> > We appreciate the suggestion. VIOLIN and relative positional encodings (RPEs) introduce spatial inductive bias through different mechanisms. As explained in Appendix B.4, VIOLIN applies a simple multiplicative decay mask, while modern RPEs add learned pairwise positional terms and often require substantial extra parameters. To address this concern directly, we expanded our experiments to include several RPE-based locality baselines in both the pretraining and finetuning settings.
> >
> > **(a) Pretraining.** On ImageNet-1K pretraining, VIOLIN achieves competitive performance to several RPE variants while adding significantly fewer FLOPs. For example, on DeiT-S, VIOLIN introduces **5$\times$ fewer FLOPs than Transformer-XL** and **1.3$\times$ fewer FLOPs** than iRPE-QK, while obtaining comparable accuracy:
> >
> > | Model          | Additional FLOPs (%) | Top-1 Acc. |
> > | -------------- | -------------------- | ---------- |
> > | Baseline       | –                    | 79.9       |
> > | **VIOLIN**     | **0.7**              | **80.7**   |
> > | Transformer-XL | 4.3                  | 80.8       |
> > | iRPE-K         | 0.9                  | 80.9       |
> > | iRPE-QK        | 2.2                  | 81.1       |
> > | iRPE-QKV       | 5.9                  | 81.4       |
> >
> > Architectures such as SP-ViT, Swin, and SwinV2 include additional components (e.g., hierarchical structures, deeper patch embeddings, position-aware CNNs), making them not directly comparable to a plain DeiT + VIOLIN setting.
> >
> > We also observe that VIOLIN can complement learned RPEs. On DeiT-T, adding VIOLIN to iRPE-K yields an additional accuracy boost, indicating that the two methods encode different inductive information:
> >
> > | Model               | Additional FLOPs (%) | Top-1 Acc. |
> > | ------------------- | -------------------- | ---------- |
> > | Baseline            | –                    | 72.2       |
> > | iRPE-K              | 1.7                  | 73.7       |
> > | **iRPE-K + VIOLIN** | 2.3                  | **73.9**   |
> >
> > **(b) Finetuning.** A key strength of VIOLIN is its **plug-and-play nature for low-data regimes**. To evaluate this, we compare several RPE-based and other locality methods in the finetuning setting on the Structured group of VTAB, where spatial inductive biases are most impactful. All methods start from the same pretrained DeiT-B, are initialized so that initial behavior remains unchanged, and are finetuned using the same VTAB hyperparameter search (Appendix G.2):
> >
> > | Method             | # Extra Params | Structured Avg. (%) |
> > | ------------------ | -------------- | ------------------- |
> > | Baseline (DeiT-B)  | –              | 57.00               |
> > | **VIOLIN (ours)**  | ~1.3K          | **61.89**           |
> > | Additive VIOLIN    | ~1.3K          | 61.34               |
> > | Swin RPB           | ~105K          | 61.58               |
> > | iRPE-QKV           | ~115K          | 61.45               |
> > | LocalViT           | ~6.2M          | 61.50               |
> > | Manhattan Mask     | ~0.4K          | 58.37               |
> > | Single SFC (Peano) | ~0.4K          | 61.63               |
> > | Random Curve       | ~0.4K          | 61.43               |
> >
> > These results show that while many locality priors (including RPEs) offer improvements, VIOLIN provides **the strongest gains with the lowest overhead** with only ~1.3K parameters compared to 10⁵–10⁶ for many RPE variants.
> >
> > All new results have been added to Section 4.3 Appendices F.6 and F.8 of the revised manuscript.

---

### Official Review · Reviewer_cmWr · 2025-10-31

**Soundness:** 3
**Presentation:** 3
**Contribution:** 2
**Rating:** 6
**Confidence:** 4

**Summary:**

The paper introduces the use of space filling curves as a way to introduce spatial priors to vision transformers. It extends upon the use of decay masks with image flattening as determined by different space filling curves. The use of different curves effectively reorders the patches of the image in different spatially meaningful ways as compared to a single zig-zag line scan used in transformer architectures. The proposed method improves upon previous data efficient methods under similar settings and can also be applied solely in the fine-tuning stage.

**Strengths:**

The authors proposed a novel way to include hand designed spatial priors thru the use of SFCs and proposed an efficient and effective way to incorporate into ViT architectures. Their proposed method can also be included into pretrained models with fine-tuning only. The proposed method improves on previous data-efficient methods like DeiT. Well designed ablation studies were also included to show the effects of each of their proposed changes to the attention mechanism. The authors also include a rather commendable and substantial appendix with important key prior art.

**Weaknesses:**

- Training flow is not immediately clear in the main paper. Since there are multiple stages to train a ViT with VIOLIN masks, it would be good to recap on the stages even though DeiT’s training recipe was followed. This would make the experiment section and the ablation studies clearer.
- The authors proposed the use of different hand selected SFCs, it would be interesting to see how a separately learned patch ordering, e.g. from Kutscher 2025, compares. After all, the mask decay method can take in any form of ordering.
- Minor issue: Typo in Figure 2\. In the center block, VIOLIN is misspelled.

**Questions:**

- DieT uses CNN as a teacher network. Is a CNN also used in this case?
- Since CNNs have the strongest spatial prior, could the authors also include a similarly size SOTA CNN? Especially if, similar to DieT training recipe, a CNN is used as a teacher network

---

> ### Author Response · Authors · 2025-11-19
>
> We thank the reviewer for the detailed evaluation of our work and for highlighting the **novel, efficient, and effective** integration of SFCs into ViTs, the strong performance in data-efficient settings, and the plug-and-play use of VIOLIN during finetuning. We also appreciate the recognition of our **ablations and detailed appendix**. Below we address the raised concerns.
>
> ---
> ### **1. Tranining flow and use of CNNs**
>
> We clarify that DeiT paper contains two components: a data-efficient training recipe and a distillation framework. We only follow the **DeiT training recipe** (i.e., data augmentation, regularization, and optimization settings). We **do not use DeiT’s distillation mechanism** and **do not use a CNN teacher** at any stage. This is now stated more clearly in the main text.
>
> Our pretraining is a single stage procedure on ImageNet-1K using original DeiT recipe without distillation. For finetuning, we rely on the [VTAB-1K repository](https://github.com/BenediktAlkin/vtab1k-pytorch/tree/main) and protocol, applied to all models (baselines and VIOLIN variants) under the same hyperparameter search described in the Appendix G.2
>
> Thanks to your suggestion, we have added similarly sized ResNet models as CNN baselines to Table 5, which clearly shows that VIOLIN significantly outperforms the mentioned CNNs.
>
> ---
> ### **2. Learned patch ordering**
>
> We appreciate the suggestion and implemented the suggested learned patch-ordering variant [1] within our framework and trained a DeiT-Tiny model using this learned ordering. The results are shown below. Although the learned version underperforms the VIOLIN mask, this opens up many interesting research directions such as how to learn multiple curves simultaneously or how different tasks result in specialized orders (which would increase interpretability). We thank the reviewer for this suggestion which helped us to broaden the scope of our analysis.
>
> | Model | Accuracy (%) |
> | -------- | -------- |
> | DeiT-T     | 72.2     |
> | VIOLIN    | **73.0**     |
> | VIOLIN  w learned order  | 70.1     |
>
> We additionally thank the reviewer for pointing the typo in Figure 2 which has now been corrected.
>
> ---
> [1] D. Kutscher et al., REOrdering Patches Improves Vision Models, 2025.

---

### Official Review · Reviewer_8zFr · 2025-11-01

**Soundness:** 3
**Presentation:** 3
**Contribution:** 2
**Rating:** 6
**Confidence:** 3

**Summary:**

The manuscript proposes a lightweight masked attention mechanism named VIOLIN that integrates Space Filling Curves (SFCs) to enhance spatial awareness in smaller visual transformers (ViT). By better filling the space in 2D images through specifically designed curves, a better neighborhood representation is achieved when applying ViTs. VIOLIN scans the input image with multiple SFCs to build curve specific decay masks which are averaged and then weighted with the attention matrix to encode spatial relationships.

As SFCs the authors use Snake, Zig-zag, Peano, and Hilbert curves together with their transposed variants to capture diverse scanning patterns in both row and column major order.

**Strengths:**

- The author propose an approach to represent better the neighbourhoods through Space filling curves (SFC) in order to enhance the processing of the image with ViT networks.
- The manuscript concludes that by using SFCs improves the performance in performance in small models and limited-data settings.
- Extensive experimental results are provided.
- The approach requires only limited extra computational demands
- Extending the application of SFCs to video understanding is also assessed.

**Weaknesses:**

- There is no systematic or any theoretical study about what the space filling curves are useful for in ViTs
- It is not clear what applications can be used for such SPCs based representations in ViTs except for some particular filtering. In the manuscript it is indicated that it can be applied for classification, semantic segmentation or object detection.
- It is not clear how such SPCs can be used to some other ViT models.

**Questions:**

Could the multiple SFC scans be combined in a more efficient way than by simply averaging?

How would the proposed multiple SFC work in the case of other transformer networks than those tested in the manuscript?
For example how would they work for the Swin transformer proposed in the paper:
Z. Liu et al., Swin Transformer: Hierarchical Vision Transformer using Shifted Windows, ICCV 2021.

How it would work, when applied on videos, on some video transformers, like for:
Limin, W. el al, VideoAME V2: Scaling}video masked autoencoders with dual masking, CVPR 2023.

---

> ### Author Response · Authors · 2025-11-19
>
> We thank the reviewer for the thoughtful evaluation of our work and for highlighting the use of multiple SFCs to enhance spatial awareness, **the strong results in small-model, limited-data settings**, and the **extensive experimental validation**. We also appreciate the recognition of VIOLIN’s **minimal overhead** and its potential extension to video understanding. Below we address the raised concerns.
>
> ---
> ### **1. Theoretical understanding and usefulness of SFCs in ViTs**
>
> Attention of ViTs is permutation invariant, and as a result, it lacks inherent spatial bias [1]. Similar to linear transformers and state-space models [2, 3, 4], the decay masks used in VIOLIN enforce local information by breaking the permutation invariance of attention, as described in Appendix C. [5] also shows how these decay masks can theoretically encode the local positions of tokens in bidirectional tasks, such as image classification.
>
> ---
> ### **2. Applications of VIOLIN**
>
> Our experiments show that VIOLIN provides the strongest benefits on tasks that **explicitly rely on spatial reasoning**, such as the Structured VTAB tasks and pixel-level CIFAR-100, where preserving the neighborhood structure is crucial. As also stated by the reviewer and shown by extensive experiments in Section 4 of our paper, beyond classification, VIOLIN improves on semantic segmentation and object detection, confirming that SFC-based priors are helpful in dense prediction tasks.
>
> In general, VIOLIN can be used in **any setting where spatial priors are important and a global attention mechanism is used**. This opens up many exciting future directions, including applications to depth estimation, super-resolution, tracking, and even video understanding.
>
> ---
> ### **3. Alternative ways of combining SFCs**
>
> For efficiency, VIOLIN mask is applied to a single $QK^{\top}$ matrix, which requires merging different SFC masks into one aggregated mask aligned with the original ordering. Therefore, some form of averaging or aggregation across curves is unavoidable.
>
> We chose simple averaging because it is **stable, adds no runtime overhead, and preserves information from all curves**. We also experimented with different combinations such as learned weighting and gating, but these added more parameters and complexity without providing meaningful performance gains. Therefore, we opted for simple and effective averaging.
>
> ---
> ### **4. Applicability to other ViT architectures**
>
> VIOLIN is most effective in models with **global attention**, where a global SFC ordering meaningfully can change interactions among all tokens. In contrast, architectures like Swin compute attention only within **local windows**, which already provide strong spatial inductive bias. As also noted in Appendix D.2, this makes SFC-based priors less suitable for these models. When applied to Swin, VIOLIN resulted in only small gains (+0.1–0.2%), which is consistent with this analysis.
>
> ---
> ### **5. Extension to video transformers**
>
> Video transformers operate on spatiotemporal tokens, and VIOLIN can be incorporated to these models in a straightforward way as it only rescales the attention scores between tokens. This makes VIOLIN orthogonal to methods such as dual masking in VideoMAE V2.
>
> There are two natural ways to extend VIOLIN to video models:
> 1. **Spatial-only SFCs (2D per frame):** It is possible to apply the same 2D SFCs used for images to the (h,w) grid of each frame, while keeping time unchanged. This encodes per-frame locality only.
> 2. **Full spatiotemporal SFCs (3D):** Following Definition 1, SFCs naturally generalize to arbitrary dimensions. This would allow us to define 3D spatiotemporal SFCs over the (t,h,w) grid (e.g., 3D Hilbert or 3D Morton curves) and compute distances based on each token’s original position. The resulting decay masks naturally encourage locality across both space and time. Masks can be computed on the full grid and then indexed to the visible token subset, exactly like positional embeddings in VideoMAE.
>
> Both ways are fully compatible with video MAE-style training since they require **no change to masking or reconstruction objectives**. They can be applied to both encoder and decoder, and they provide a meaningful structural prior, especially under high masking ratios, where positional structure becomes crucial. Overall, extending VIOLIN to video models is a promising direction for future work, as it can introduce strong spatial–temporal inductive bias with minimal additional cost.
>
> ---
> [1] A. Dosovitskiy et al., An Image is Worth 16×16 Words: Transformers for Image Recognition at Scale, 2020.
>
> [2] A. Gu and T. Dao, Mamba: Linear-Time Sequence Modeling with Selective State Spaces, 2023.
>
> [3] Y. Sun et al., Retentive Network: A Successor to Transformer for Large Language Models, 2023.
>
> [4] Z. Qin et al., HGRN2: Gated Linear RNNs with State Expansion, 2023.
>
> [5] A. Afzal et al., Linear Attention for Efficient Bidirectional Sequence Modeling, 2024.

---

### Meta-Review · Area_Chair_mYUW · 2026-01-09

**Summary:**

This paper introduces VIOLIN, a lightweight masked attention mechanism that integrates Space Filling Curves to inject spatial inductive biases into Vision Transformers, targeting small-model and data-scarce regimes. The reviews highlighted strengths such as the method's plug-and-play design, negligible computational overhead, and strong empirical gains on typical tasks. However, reviewers initially raised concerns about novelty, limited comparisons to relative positional encodings, and generalization to architectures like Swin.

The authors addressed these points in the rebuttal: they added direct comparisons to RPE baselines, showing VIOLIN achieves comparable performance with fewer parameters; they clarified that VIOLIN's gains are most pronounced in global-attention models (e.g., DeiT) rather than locally constrained ones like Swin; and they provided theoretical justifications for mask averaging. These responses strengthen the paper's contributions, particularly in validating VIOLIN’s efficiency and scope of applicability.


It is noteworthy, however, that a reference appears to be hallucinated, for example:

"I. J. Schoenberg. A remark on the mapping of the line onto a plane region. Bulletin of the American Mathematical Society, 44(12):831-834, 1938."

Given the presence of the hallucinated reference, it is recommended that this submission be desk-rejected.

**Reviewer Concerns:**

See above.

**Reviewer Scores:**

Reviewers might change their ratings to positive ones after rebuttal if they overlook the hallucinated references.

---

### Decision · Program_Chairs · 2026-01-26

Reject